# Analytical model captures intratidal variation of salinity in a convergent, well-mixed estuary

Yanwen Xu[1, 2, 3], Antonius J. F. Hoitink[4], Jinhai Zheng[2, 5], Karl Kästner[4], Wei Zhang[2, 5]

[1] Key Laboratory of Coastal Disasters and Defense of Ministry of Education, Hohai University, Nanjing 210098, China
[2] College of Harbor, Coastal and Offshore Engineering, Hohai University, Nanjing 210098, China
[3] Key Laboratory of the Pearl River Estuarine Dynamics and Associated Process Regulation, Ministry of Water Resources, Guangzhou 510611, China
[4] Hydrology and Quantitative, Water Management Group, Department of Environmental Sciences, Wageningen University, The Netherlands
[5] State key laboratory of Hydrology-Water Resources and Hydraulic Engineering, Hohai University, Nanjing 210098, China

*Correspondence to*: Wei Zhang (zhangweihhu@126.com)

**Abstract.** Knowledge of the processes governing salt intrusion in estuaries is important since it influences the eco-environment of estuaries as well as its water resource potential in many ways. Analytical models of salinity variation offer a simple and efficient method to study salt intrusion in estuaries. In this paper, an unsteady analytical solution is presented to predict the spatial-temporal variation of salinity in convergent estuaries. It is derived from a one-dimensional advection-diffusion equation for salinity, adopting a constant mixing coefficient and a single-frequency tidal wave, which can directly reflect the influence of the tidal motion and the interaction between the tide and runoff. The deduced analytical solution is illustrated with an application to the Humen estuary of the Pearl River Delta (PRD), and proves to be an efficient and accurate approach to predict the salt intrusion in convergent estuaries. The unsteady analytical solution is tested against observations from six study sites, to validate its capability to predict intratidal variation of salt intrusion. The results show that the proposed unsteady analytical solution can be successfully used to reproduce the spatial distribution and temporal processes governing salinity dynamics in convergent, well-mixed estuaries. The proposed method provides a quick and convenient approach to decide upon water fetching works to make good use of water resources.

## 1 Introduction

Salt intrusion in a river communicating with the sea is largely controlled by the river flow (Keulegan, 1966). The salinity of estuary waters is the result of the balance between river and tidal fluxes, and mixing between them. The natural variability of river and tidal inputs to estuaries has been greatly disrupted as a result of the impact of global climate change and sea level rise as well as of local human activities, such as dam construction and channel dredging. These changes cause salt intrusion to become a serious problem in estuaries. It influences water quality, agricultural development in lowland areas, water utilization in upstream catchments, and the aquatic environment in estuaries (Han et al., 2010; Mo et al., 2007; Savenije, 1992). To address

this issue worldwide, research efforts devoted to salt intrusion have been conducted in laboratory tanks, with numerical models and using analytical approaches.

Nowadays, numerical models have become the most popular tool to study salinity distribution in estuaries, because they can provide visible results presenting the spatial-temporal variation in detail (e.g. Gong et al., 2012; Lerczak et al., 2006; MacCready, 2004; Wu and Zhu, 2010). However, the application of a numerical model is not an easy task since it requires detailed data of the bathymetry, and of hydrological boundary conditions, which are not available for all estuaries in the world. Here, a comparatively simple and convenient analytical model is developed as an efficient method to study the salt intrusion in well-mixed estuaries. Analytical models are widely used because they are simple, while retaining the basic physical characteristics involved. In the early 1960s, when systematic methods were developed to explore the factors controlling the instantaneous longitudinal salinity distribution, an expression was developed to compute the salt intrusion length as a function of the estuary length, mean depth, tidal amplitude, tidal period, fresh-water discharge, ocean salinity, and estuary roughness (Ippen and Harleman, 1961). In a subsequent period, analytical models of increasing complexity were developed based on the one-dimensional advection diffusion equation (Cameron and Pritchard, 1963), and on two-dimensional equations (Hansen and Rattray, 1965), capturing the dynamics of buoyancy driven exchange flow and tidal mixing, satisfying salt conservation. Since the 1970s, numerous empirical and semi-empirical one-dimensional analytical models were put forward that correlated the salt intrusion length to the estuarine dynamical conditions and geomorphology, based on the flume experiments and field measurements (e.g. Brockway et al., 2006; Fischer, 1974; Gay and O'Donnell, 2007, 2009; Kuijper and Rijn, 2011; Lewis and Uncles, 2003; Prandle, 1981, 1985; Rigter, 1973; Savenije, 1986). Although the literature on salt intrusion is vast, most studies concentrate on salt water intrusion in a prismatic flume for reasons of convenience. However, the majority of estuaries in the world converge in width. The topography of the estuary is crucial to salt intrusion, because the two main drivers (i.e. river flow and the tidal motion) both depend on the topography. The cross-section area determines the amount of the salt water entering the estuary, and the efficiency of fresh water carrying salt out of the estuary. Savenije (1986A) developed a fully analytical and predictive model to predict salt intrusion that applies to the natural topography of alluvial estuaries. It has been well validated in numerous estuaries where the width converges exponentially (e.g. Savenije, 1989; Savenije and Pagès, 1992; Nguyen and Savenije, 2006; Eaton, 2007; Ervine et al., 2007; Nguyen et al., 2008). In the years 2000-2010, another analytical approach (Brockway et al., 2006) was put forward, which can be considered as a modified and simplified version of the method presented in earlier studies (Prandle, 1981; Savenije, 1986). The dispersion coefficient in Brockway's model is assumed to be constant along the estuary, while it is assumed to be proportional to the spatial integral of the subtidal axial velocity in Savenije's model. In the theoretical models described above, the salt intrusion is predicted as a steady-state solution during slack water. Few studies have focused on analyzing the intratidal variation of salinity analytically. Song et al. (2008) proposed an unsteady-state model applicable to laboratory flumes and artificial channels where the cross section is assumed to be constant along the channel. Elaborating on the work of Song et al. (2008), here, an unsteady-state model is developed to predict the intratidal salinity intrusion dynamics in alluvial estuaries where the cross-section area typically converges. The aim of this

study is to introduce a simple, unsteady analytical solution to the problem of predicting the intratidal variation of salt intrusion in convergent, well-mixed estuaries.

## 2 Methods

The cross-sectional area in this paper is described as an exponential function:

$$A = A_0 \exp(-x/a), \tag{1}$$

where $A_0$ is the cross-sectional area at the mouth ($x$=0), $x$ is distance along the estuary and $a$ is the convergence length of the cross-sectional area. The $x$-axis has its origin at the mouth of the estuary and the upstream direction is taken as positive. The one-dimensional advection-diffusion equation for salinity can be written as follows:

$$\frac{\partial As}{\partial t} + \frac{\partial Aus}{\partial x} = \frac{\partial}{\partial x}\left(AD\frac{\partial s}{\partial x}\right), \tag{2}$$

where $s$ is salinity averaged over the cross-section, $t$ is time, $u$ is the velocity and $D$ is the longitudinal dispersion coefficient. Although the assumption of a variable coefficient seems to be more reasonable, models with a constant dispersion coefficient have also proved to be capable of satisfactorily reproducing the salinity distribution (Lewis and Uncles, 2003; Brockway et al., 2006; Gay and O'Donnell, 2007, 2009). Under the assumption that $D$ is independent of time, the salinity can be expanded in a Fourier series and be expressed as (Song et al., 2008):

$$s = \bar{s} + s_1 \cos(\omega t + \varphi) + s_2 \sin(\omega t + \varphi), \tag{3}$$

where $\bar{s}$ is the tide-averaged salinity, $s_1$ and $s_2$ are coefficients. For the case of a simple harmonic wave with river discharge, the instantaneous flow velocity $u$ is considered to consist of a time-dependent component $u_t = \upsilon\cos(\omega t + \varphi)$ created by the tide, and a steady component $u_f = Q_f/A$ contributed by the river flow:

$$u = u_f + \upsilon\cos(\omega t + \varphi), \tag{4}$$

where the value of the runoff velocity $u_f$ is negative. Introducing Eqs. (3) and (4) into Eq. (2), and using $\theta = \omega t + \varphi$ yields:

$$\left(u_f \frac{\partial s_2}{\partial x} - \omega s_1\right)\sin\theta + \left(\omega s_2 + u_f \frac{\partial s_1}{\partial x} + \upsilon\frac{\partial \bar{s}}{\partial x}\right)\cos\theta + u_f \frac{\partial \bar{s}}{\partial x} + \frac{\upsilon}{2}\frac{\partial s_1}{\partial x}$$
$$= \left(D\frac{\partial^2 s_2}{\partial x^2} - \frac{D}{a}\frac{\partial s_2}{\partial x}\right)\sin\theta + \left(D\frac{\partial^2 s_1}{\partial x^2} - \frac{D}{a}\frac{\partial s_1}{\partial x}\right)\cos\theta + D\frac{\partial^2 \bar{s}}{\partial x^2} - \frac{D}{a}\frac{\partial \bar{s}}{\partial x}, \tag{5}$$

As the equation should hold for all values of $\theta$, Eq. (5) yields the following set of equations:

$$\begin{cases} -\omega s_1 + u_f \dfrac{\partial s_2}{\partial x} = D \dfrac{\partial^2 s_2}{\partial x^2} - \dfrac{D}{a}\dfrac{\partial s_2}{\partial x}, \\[2mm] \omega s_2 + u_f \dfrac{\partial s_1}{\partial x} + \upsilon \dfrac{\partial \bar{s}}{\partial x} = D \dfrac{\partial^2 s_1}{\partial x^2} - \dfrac{D}{a}\dfrac{\partial s_1}{\partial x}, \\[2mm] u_f \dfrac{\partial \bar{s}}{\partial x} + \dfrac{\upsilon}{2}\dfrac{\partial s_1}{\partial x} = D \dfrac{\partial^2 \bar{s}}{\partial x^2} - \dfrac{D}{a}\dfrac{\partial \bar{s}}{\partial x}, \end{cases} \tag{6}$$

where $s$, $s_1$ and $s_2$ can be further assumed as:

$$\bar{s} = c_0 \exp\left( m\left( \exp\left(\frac{x}{a}\right) - 1 \right) \right),$$

$$s_1 = c_1 \exp\left( m\left( \exp\left(\frac{x}{a}\right) - 1 \right) \right), \tag{7}$$

5  $$s_2 = c_2 \exp\left( m\left( \exp\left(\frac{x}{a}\right) - 1 \right) \right),$$

with $c_0 = \bar{s}_0$ is the tide-averaged salinity at the mouth of estuary. Substitution of the equation set (7) into the equation set (6) yields:

$$\begin{cases} -\omega s_1 + \left( u_f + \dfrac{D}{a} \right)\dfrac{m}{a}\exp\left(\dfrac{x}{a}\right) s_2 = D\dfrac{m}{a}\exp\left(\dfrac{x}{a}\right)\left(\dfrac{1}{a} + \dfrac{m}{a}\exp\left(\dfrac{x}{a}\right)\right) s_2, \\[2mm] \omega s_2 + \upsilon\dfrac{m}{a}\exp\left(\dfrac{x}{a}\right)\bar{s} + \left( u_f + \dfrac{D}{a} \right)\dfrac{m}{a}\exp\left(\dfrac{x}{a}\right) s_1 = D\dfrac{m}{a}\exp\left(\dfrac{x}{a}\right)\left(\dfrac{1}{a} + \dfrac{m}{a}\exp\left(\dfrac{x}{a}\right)\right) s_1, \\[2mm] \left( u_f + \dfrac{D}{a} \right)\dfrac{m}{a}\exp\left(\dfrac{x}{a}\right)\bar{s} + \dfrac{\upsilon}{2}\dfrac{m}{a}\exp\left(\dfrac{x}{a}\right) s_1 = D\dfrac{m}{a}\exp\left(\dfrac{x}{a}\right)\left(\dfrac{1}{a} + \dfrac{m}{a}\exp\left(\dfrac{x}{a}\right)\right)\bar{s}. \end{cases} \tag{8}$$

Then, further elaboration yields:

$$\begin{cases} -\omega c_1 + \left( u_f - \dfrac{Dm}{a}\exp\left(\dfrac{x}{a}\right) \right)\dfrac{m}{a}\exp\left(\dfrac{x}{a}\right) c_2 = 0, \\[2mm] \omega c_2 + \upsilon\dfrac{m}{a}\exp\left(\dfrac{x}{a}\right) c_0 + \left( u_f - \dfrac{Dm}{a}\exp\left(\dfrac{x}{a}\right) \right)\dfrac{m}{a}\exp\left(\dfrac{x}{a}\right) c_1 = 0, \\[2mm] \left( u_f - \dfrac{Dm}{a}\exp\left(\dfrac{x}{a}\right) \right)\dfrac{m}{a}\exp\left(\dfrac{x}{a}\right) c_0 + \dfrac{\upsilon}{2}\dfrac{m}{a}\exp\left(\dfrac{x}{a}\right) c_1 = 0. \end{cases} \tag{9}$$

Hence, the solutions can be obtained:

$$\begin{cases} c_1 = \dfrac{\left(u_f - \dfrac{Dm}{a}\exp\left(\dfrac{x}{a}\right)\right)\dfrac{m}{a}\exp\left(\dfrac{x}{a}\right)}{\omega} c_2, \\[4ex] c_2 = \dfrac{-\upsilon\dfrac{m}{a}\exp\left(\dfrac{x}{a}\right)\omega}{\omega^2 + \left(u_f - \dfrac{Dm}{a}\exp\left(\dfrac{x}{a}\right)\right)^2 \dfrac{m^2}{a^2}\exp\left(\dfrac{2x}{a}\right)} c_0, \\[4ex] m = \dfrac{u_f a \exp\left(-\dfrac{x}{a}\right)}{D} = \dfrac{Q_f a}{DA_0}. \end{cases} \tag{10}$$

The analytic solution of the unsteady state salinity distribution is therefore represented as:

$$s = \bar{s}_0 \exp\left(\frac{Q_f a}{DA_0}\left(\exp\left(\frac{x}{a}\right)-1\right)\right)\left(1 - \frac{\upsilon u_f}{\omega D}\sin(\omega t + \varphi)\right). \tag{11}$$

By integrating this unsteady salinity expression over the tidal period $T$, the salt intrusion under Tidal Average conditions (TA), as defined by Brockway et al. (2006), can be obtained as:

$$\bar{s}_x = \bar{s}_0 \exp\left(\frac{Q_f a}{DA_0}\left(\exp\left(\frac{x}{a}\right)-1\right)\right). \tag{12}$$

A graph of the logarithm of salinity $\ln\left(\bar{s}/\bar{s}_0\right)$ against $\exp(x/a)$ should be a straight line, with the slope inversely proportional to the longitudinal dispersion coefficient $D$ (Brockway et al., 2006). The coefficient $D$ can then be calculated from:

$$D = \frac{Q_f a}{kA_0}, \tag{13}$$

where $k$ is the slope of the fitted line. This approach makes it possible to estimate the longitudinal dispersion coefficient $D$ based on the measurements of salinity made during a survey.

The tidal velocity amplitude $\upsilon$ can be estimated as $\upsilon = E\pi/T$ (Savenije, 1993) where $E$ is the tidal excursion and the harmonic constant $\omega$ is given as $\omega = 2\pi/T$. Introducing $\upsilon = E\pi/T$ and $\omega = 2\pi/T$ into Eq. (11), and using $u_f = Q_f/A$ yields:

$$s = \bar{s}_0 \exp\left(\frac{Q_f a}{DA_0}\left(\exp\left(\frac{x}{a}\right)-1\right)\right)\left(1 + \frac{E}{2a}\left(-\frac{Q_f a}{DA_0}\exp\left(\frac{x}{a}\right)\right)\sin(\omega t + \varphi)\right). \tag{14}$$

This expression can be used to describe the temporal and spatial variation of salinity, including High Water Slack (HWS) and Low Water Slack (LWS), when the tidal discharge is zero by definition. Since the maximum salinity is reached at HWS and the minimum salinity is reached at LWS (Savenije, 2005), Eq. (14) can be simplified for HWS into:

$$s_{max} = \bar{s}_0 \exp\left(\frac{Q_f a}{DA_0}\left(\exp\left(\frac{x}{a}\right)-1\right)\right)\left(1 + \frac{E}{2a}\left(-\frac{Q_f a}{DA_0}\exp\left(\frac{x}{a}\right)\right)\right), \tag{15}$$

and for LWS into:

$$s_{\min} = \bar{s}_0 \exp\left(\frac{Q_f a}{DA_0}\left(\exp\left(\frac{x}{a}\right)-1\right)\right)\left(1 - \frac{E}{2a}\left(-\frac{Q_f a}{DA_0}\exp\left(\frac{x}{a}\right)\right)\right). \tag{16}$$

The tidal excursion $E$, the distance over which a water particle travels up and down the estuary with the flooding and ebbing tide, is assumed to decrease exponentially along the channel:

$$E = E_0 \exp(-x/e), \tag{17}$$

where $E_0$ is the tidal excursion at the mouth ($x=0$), and $e$ is the damping length of the tidal excursion. Thus, combination of Eqs. (15), (16) and (17) yields:

$$E = \frac{a(s_{\max 0} - s_{\min 0})}{\bar{s}_0\left(-\frac{Q_f a}{DA_0}\right)}\exp(-x/e), \tag{18}$$

where $s_{\max 0}$ is the maximum salinity at the estuary mouth and $s_{\min 0}$ is the minimum salinity at the estuary mouth.

Since the tidal flow is assumed to vary as a simple harmonic wave, the unsteady salinity model is here presented in its simplest form, with a single frequency. As the tidal propagation celerity in the estuary is assumed to be constant, the tidal phase at each site can be made up of an initial phase $\varphi_0$ at the mouth of the estuary and a phase difference that is the travel time of the tide from the mouth to the study site. Therefore, Eq. (14) can be modified as:

$$s = \bar{s}_0 \exp\left(\frac{Q_f a}{DA_0}\left(\exp\left(\frac{x}{a}\right)-1\right)\right)\left(1 + \left(-\frac{E_0}{2a}\frac{Q_f a}{DA_0}\right)\exp\left(\frac{x}{a}-\frac{x}{e}\right)\sin\left(\omega\left(t-\frac{x}{c}\right)+\varphi_0\right)\right), \tag{19}$$

where $c$ is the tidal propagation celerity.

We note that in the approach presented above the tidal excursion at the mouth is inferred from salinity data, whereas an alternative theoretical approach may be applicable that is less dependent on in situ data. Tidal wave propagation can be described analytically by a set of four implicit equations (Cai et al., 2012), the phase lag equation $\tan(\varepsilon) = \lambda/(\gamma-\delta)$, the scaling equation $\mu = \sin(\varepsilon)/\lambda$, the damping equation $\delta = \gamma/2 - 4\chi\mu/(9\pi\lambda) - \chi\mu^2/3$, and the celerity equation $\lambda^2 = 1 - \delta(\lambda-\delta)$, where $\lambda$ is the celerity number $\lambda = c_0/c$, $\mu$ is the velocity number $\mu = \upsilon\bar{h}/(r_s\eta c_0)$, $\delta$ is the damping number $\delta = c_0 d\eta/(\eta dx\omega)$, and $\varepsilon$ is the phase lag between HW and HWS $\varepsilon = \pi/2 - (\phi_Z - \phi_U)$. Here three dimensionless parameters control the tidal hydrodynamics (Savenije et al., 2008), i.e. the dimensionless tidal amplitude $\zeta = \eta/\bar{h}$, the estuary shape number $\gamma = c_0/(\omega a)$ and the friction number $\chi = r_s g c_0 \zeta/(K_s^2 \omega \bar{h}^{4/3})\left[1-(4\zeta/3)^2\right]^{-1}$, where $\eta$ is the tidal amplitude, $K_S$ is the Manning-Strickler friction coefficient, $r_s$ the storage width ratio, $\bar{h}$ is the tide-averaged depth and $c_0$ is the classical wave celerity $c_0 = \sqrt{g\bar{h}/r_s}$.

Then with available geometry and friction data at the estuary mouth, the tidal propagation celerity and the tidal amplitude (or

the tidal excursion) can be obtained by solving the set of four equations. Rather than proceeding with this analytical model for tidal hydodynamics, hereafter we employ Eq. 18 to close the set of equations.

## 3 Study area and data

### 3.1 Overview of study area

The Pearl River estuary (PRE) is located midway along the northern boundary of the South China Sea. It receives a large amount of fresh water from the Pearl River which has three major branches (i.e. the West River, the North River and the East River) in the upper drainage basin. The annual river discharge, with 80% occurring in the wet season, empties into the South China Sea via eight outlets (Zhao, 1990). The Lingding Bay is created by the inflows of freshwater from the Pearl River through four major discharge outlets, namely Humen, Jiaomen, Hongqimen, Hengmen. Historically, about 50-55% of the river

flow enters the Lingding bay, while the remaining freshwater directly flows into the South China Sea through the four southwestern outlets (i.e. Modaomen, Jitimen, Hutiaomen and Yamen).

The Humen is the largest river outlet in the Lingding bay and contributes 34.6% of the water discharge, i.e. about $603 \times 10^8$ m$^3$ in terms of annual water discharge (Ren et al., 2006). The freshwater input into Lingding bay through the Humen outlet comes from three sources: the East River, the Liuxi River and the North River. The annual river discharge with a peak of 1 870 m$^3$s$^{-1}$

$^1$, measured at Niuxinling station in Liuxi River, is about 10 times less than that with a flood peak in excess of 12 000 m$^3$s$^{-1}$, measured in the other two rivers (Luo et al., 2002).

The tide in the Pearl River estuary has a mixed semidiurnal-diurnal character (Zhang et al., 2012). Among the eight outlets of the Pearl River estuary, Humen is most strongly dominated by the tide, with an annual average and maximum tidal range of 1.63 m and 2.59 m, respectively, at the mouth of the estuary (Li and Lei, 1998).

As a major tributary of the Pearl River, the Humen estuary can be divided into two waterways: the Guangzhou channel (the upper reach) with an average width of 431 m, and the Shiziyang channel (the lower reach) which is about 2200 m wide (Mai et al., 2001). It is a NW-SE branch of the Pearl River estuary with a width of about 4 km at the mouth, resembling an inverted funnel with a narrow neck in the north and a wide mouth opening to the south. The Humen outlet has the highest tidal prism in the Pearl River estuary due to the large-width of the mouth, resulting in a strong tidal motion. Especially, during spring tide

in the dry season, when the river discharge is lowest, the downstream area becomes saline.

### 3.2 Data

The information available for the model application in this study includes data on topography, salinity, river discharge and on the tidal flow. A field survey for salt intrusion was conducted during the dry season in 2005. It was a project carried out by Guangdong Province Hydrology Bureau and the Pearl Hydrology Bureau from the River Conservancy Commission. In this

paper, the field data from 29 January to 3 February were used, which were measured at six gauge stations along the channel (Figure 1). Considering the impact of shipping, the measuring positions were near the banks, with certain distances ranging

from 605 m to 70 m. A Global Positioning System was used to confirm the exact measuring locations (Table 1). The Humen estuary is well-mixed under normal flow conditions during the dry season (Ou, 2009; Luo et al., 2010). Due to three years of drought, the river discharge decreased by 50 percent during the study period in 2005 compared to a normal year (Liao, Pan, and Dong, 2008). Thus, there is no doubt that well-mixed conditions prevailed during the calibration and validation. The average salinity of vertical profiles was calculated based on the hourly water samples. At each location, the saline water was sampled at two different elevations: at 1/5 and 4/5 of the depth of channel from the bed, and salinity was obtained using a salimeter. The water discharge at stations were provided by the Hydrology Bureaus during the field survey. The cross-section was measured at mean sealevel, with the help of an ultrasonic echo-sounder.

Because of the complex river network upstream of the Humen area in the Pearl River estuary, the river discharge is difficult to determine. The total flux through the Humen outlet is composed of three parts which come from three main sources: the East River, the North River and the Liuxi River. The river discharge used in this paper was measured at upstream stations (Sanshui for the North River; Boluo for the East River and Laoyagang for the Liuxi River) from 29 January to 3 February. These data were collected from the official databases of the Hydrology Bureaus mentioned above. In the lower reach of the East River and the Liuxi River, respectively, Boluo station and Laoyagang station are located about 80 km upstream from the Humen outlet. The daily discharge measured at the Boluo station varied from 260 $m^3s^{-1}$ to 400 $m^3s^{-1}$ during the survey period, while it was about 20 $m^3s^{-1}$ at the Laoyagang station. The discharge of the East River and the Liuxi River entirely flow toward the Chinese sea through the Humen estuary (Ren et al., 2011). The North River is an important source for the river discharge to the Humen outlet. River discharge from the North River reaches the Humen channel through a network of channels which connects to the western part of the Pearl River delta. Sanshui station is the primary hydrological station in the North River. About 11% of the measured discharge flows into the Humen estuary during the survey in 2005. Shanshui station is located further upstream than the other two stations (Boluo and Layaogang). The response lag of salinity variation at the estuary mouth to discharge variation at Sanshui station is about two days, while the river flow spends about one day to travel from Boluo and Laoyagang station to the estuary mouth.

## 4 Results

### 4.1 Model calibration

To demonstrate the practical application of the proposed analytical solution, the model has been used to simulate and analyze the spatial-temporal variation of salt intrusion in the Humen Estuary. In the following, the parameters of the analytical solution are obtained from calibration.

The spatial decay of the cross-sectional area of the Humen estuary can be described by the exponential function expressed in Eq. (1). The field data (triangles) and the best-fit line are shown in Figure 2. The cross-sectional area at the mouth at mean tide, $A_0$, is calculated as 37 822 $m^2$ and the convergence length of cross section $a$ is obtained by curve fitting as 16.7 km.

The relative salinity is plotted as $\ln(s/s_0)$ against $\exp(x/a)$ in Figure 3. There is a straight line fit between these two variables, confirming the constancy of the dispersion coefficient. The dispersion coefficient $D$ can be computed from the slope of the fitted lines according to Eq. (12), where $k$ is the slope. This approach has shown previously to be efficient (e.g. Brockway et al., 2006; Fang et al., 2006; Zhang et al., 2010). Table 2 shows the results of the fit for all these surveys carried out between

29 January to 3 February (the slope in column 4 and the dispersion coefficient in column 6). Dispersion coefficient estimates obtained from this fitting procedure can be interpreted as a spatial average, representing the entire reach. The coefficient of determination ($R^2$) lies in the range between 0.85 and 0.92, with a mean value of 0.89. The assumption of the dispersion coefficient independent of distance is demonstrated to be reasonable and acceptable in the present case. The dispersion coefficient from data on 29 January is therefore used as the calibrated value.

The tidal excursion at the mouth of the estuary is obtained through Eq. (18). For each tidal excursion at each day, the period-averaged value, maximum value and minimum value of salinity at the mouth are obtained by statistical analysis, and the longitudinal dispersion coefficient $D$ is computed by linear fitting, as shown previously. Moreover, the damping length of the tidal excursion $e$ is calibrated using the observed salinity along the estuary. Similar to the tidal excursion, the value of the propagation celerity $c$ is also obtained by calibration based on observations. The initial tidal phase $\varphi_0$ is calculated via a reverse

procedure by calibration on the salinity at the mouth of estuary. Data from 31 January is used to verify the change of salinity over a tidal cycle.

The five calibration parameters (i.e. the convergence length of cross section $a$, the dispersion coefficient $D$, the tidal excursion $E_0$, the damping length of the tidal excursion $e$, the initial phase $\varphi_0$ and the tidal celerity $c$) are obtained based on the measurements at the mouth of the estuary, as shown in Table 3. Based on the observed data on 29 January, the results of the

model calibration can be seen in Figure 4.

## 4.2. Model Validation

A validation of the unsteady model is offered in two separate parts, i.e. the longitudinal distribution of salinity along the channel and the temporal variation of salinity during the tidal period. In the first part, observations during two characteristic conditions (i.e. HWS and LWS) are chosen to validate against the calculated results of the salinity distribution. In the second

part, a model for expressing the change process of salinity during tidal periods is established, according to the measurement on 31 January.

### 4.2.1. Longitudinal distribution of salinity

Based on the field measurements from 30 January to 3 February, Eqs. (15), (16) and (18) are used to calculate the longitudinal variation of salinity. Conditions of neap tide are considered to last from 31 January to 2 February. The calibration results are

presented in Figure 5. The good agreement between the computation and the measured data indicates that the performance of the unsteady analytical model is to a certain extent satisfactory in Humen estuary. The analytical model is found to better

reproduce the distribution of salinity at high water (HW) than at low water (LW). This can be attributed to different degrees of mixing, which is stronger at HW. As the estuary is assumed to be well-mixed, the analytical model undoubtedly will perform better when mixing is higher. Fluctuations around the theoretical curve may partly be caused by the unequal distribution of salinity over the cross-section, or by the indirect derivation of the salinity at HWS and LWS, which is replaced with the daily maximum and minimum values, respectively.

It can be seen that the analytical model substantially overestimates the salinity in the downstream part of the estuary, partly because of the special locations of the stations (some are located at the confluence of rivers). The expression for the distribution analysis of salinity, Eq. (11), is multiplied the tidal average salinity with an extra component that reflects the effect of the tide and the interaction of the tide and river flow. This time-dependent component is a sine function, viz. $\left(-\upsilon u_f/\omega D\right)\sin(\omega t+\varphi)$, thus the calculated salinity at HWS and LWS is always symmetrical about the average values. The symmetry property of salinity has been demonstrated by Savenije (1989) under the assumption that the tidal excursion is independent of distance. After a transformation of variables, the sine function mentioned above is expressed as $\left(E/2a\right)(-Q_f a/DA)\sin(\omega t+\varphi)$ where the dispersion coefficient plays an important role. To simplify and clarify the interaction between the tidal motion and the river flow, the parameters $D$ and $Q_f$ are combined into one single calibration variable, the mixing coefficient $\alpha$:

$$\alpha=-D/Q_f, \tag{20}$$

which can be obtained in the same way as the dispersion coefficient. In this paper, the mixing coefficient is assumed constant along the channel, to develop a comparatively simple analytical solution within acceptable levels. It is calibrated by the measurements from Dahu station to Huangpuyou station, located at the junction of two reaches in the Humen estuary.

**4.2.2. Periodic variation of salinity**

The observations of salinity at hourly intervals along the Humen estuary are used to calibrate the dispersion coefficient in the model, and to analyze the change of salinity with time. The results indicate that the calibrated unsteady analytical model fits the observations well. Figure 6, where the analytical solution is compared with observation, demonstrates that the proposed unsteady analytical solution is able to reflect the change process of salinity over a tidal cycle. Additionally, the simplification and assumption of the tidal celerity ($c$) and the initial phase at the mouth of estuary ($\varphi_0$) in Eq. (19) proves realistic.

The theoretical result of the periodic variation of salinity is not always consistent with the observations. As can be seen in Figure 6, the analytical model for simulating the temporal process of salinity has a relatively poor performance at the sites near the mouth of estuary, such as Dahu station. By comparing the variation of salinity at different sites (Figure 6), it shows that salinity variation is more symmetrical further away from the study site. The discrepancies near the mouth may have three reasons. Firstly, lateral residual circulation usually exists at the mouth of an estuary, where the cross section is widest. Secondly, the mouth of estuary is close to Lingding Bay, where the salt dynamics are influenced by coastal and ocean currents. Thirdly, near the outlet, comprehensive salinity measurements are much more difficult to take, due to the impact of tidal flats and

complex hydrodynamics, influenced by Coriolis forcing and wind effects. All the influences above are related to the width of the channel, which gradually decreases in the landward direction.

The observations at Machong station show some nonperiodic variation, which may relate to the proximity of the confluence of the East River and the Shiziyang channel. At Dasheng station, about 2.6 km upstream from Machong station and near another confluence, the simulated temporal process of salinity shows a fairly good agreement with the observations. To understand the irregular changes of salinity at Machong station, the daily averaged discharges at Machong and Dasheng stations are analyzed by integrating over the tidal period. The results are presented in Figure 7 where the positive values represent the mean discharge transporting in the seaward direction. At Machong station, the mean discharge is directed inland, which can be attributed to Stokes transport (Buschman et al., 2010; Hoitink and Jay, 2016). At Dasheng station, only a few kilometers upstream, the mean discharge is seaward, as expected. The tide-averaged discharge thus converges in the estuary during low river flow, which will increase the total water volume in the estuary, and create a mean water level rise. We expect this process has an impact on the mean salt balance, which explains part of the observed discrepancies.

For comparison, the result obtained by Song's model is also presented here. The unsteady analytical model developed by Song et al. (2008) can reproduce the salinity process in an idealized estuary with constant depth and constant width, which is expressed as:

$$s = \overline{s}_0 \exp\left(\frac{u_f}{D_S} x\right)\left(1 - \frac{u_f \upsilon}{D_S \omega} \sin\left(\omega t + \varphi\right)\right). \tag{21}$$

Therefore, in fact, it is more suitable for use in prismatic channels. The dispersion coefficient of Song's model is assumed to be independent on the distance, and can be estimated by:

$$D_S = -\frac{\overline{s} u_f \upsilon}{0.5\omega\left(s_{max} - s_{min}\right)}. \tag{22}$$

When an estimation for tidal velocity is made according to the relation $\upsilon = E\pi/T$, and the value of the runoff velocity is obtained using $u_f = Q_f/A$, then the dispersion coefficient can be calculated based on the measured salinity at the mouth of the estuary. The data which has been used for modelling in Humen estuary can be found in Table 4. As shown in Figure 8, the performance of Song's model for the Humen estuary is satisfactory at the study sites close to the estuary mouth, e.g. the Dahu station and the Sishengwei station. However, the salt intrusion is underestimated by the model at the Zhangpeng station (Figure 8c) and the Dasheng station (Figure 8d), in the upstream part of the estuary. A likely reason for the underestimation can be the fundamental assumption that the channel has a constant cross section. The river width convergence at the Humen estuary can actually be described by an exponential function. Simplifying this estuary geometry can result in the underestimation of the mixing coefficient. It indicates that topography is a key driver of the salt intrusion along the Humen estuary.

### 4.3. Sensitivity analysis

The amplitude of salinity can be described by:

$$\hat{s} = \overline{s}_x * I_s, \tag{23}$$

where $\overline{s}_x$ is the tide-averaged salinity along the estuary and is a function of the river discharge, i.e. Eq.(12). The parameter $I_s$ is the salinity amplitude coefficient that is defined as:

$$I_s = -\frac{EQ_f}{2DA}, \tag{24}$$

representing the interaction between tides and the river discharge. To investigate the longitudinal salinity distribution and intratidal salinity variation for different discharge and tidal dynamic conditions in the Humen estuary, Eqs. (12) and (23) are used to plot the longitudinal salinity curve and intratidal variation of salinity, respectively. The implemented parameters are the same as shown in Table 3, only the river discharge and the tidal excursion are variable.

Three constant discharge values of 200, 600 and 1800 m³/s are used to evaluate the impact of the river discharge on the salinity variation. The discharge values are chosen because the minimum discharge in the dry season is around 600 m³/s in the Humen estuary, and low salinity can be measured at Huangpuyou station when the discharge is larger than 1800 m³/s. In addition, the discharge in the extreme dry season is set to be 200 m³/s. The longitudinal salinity curve can be seen Figure 9a. At tidal average conditions, the salt intrusion length becomes smaller when the discharge increases. The steepest salinity gradient can be found at the highest discharge ($Q_f$=1800 m³/s). It is clear from Figure 9b that the salinity amplitude increases firstly and then decreases as the river discharge increases. This is because during periods of low river discharge ($Q_f$= 200 m³/s), the tide-averaged salinity is larger but the salinity amplitude coefficient $I_s$ is smaller, which indicates the weaker interaction between the river flow and the tides. However, the tide-averaged salinity decreases rapidly with the increasing river discharge, as we can see from Figure 9a, resulting in a smaller amplitude of salinity during periods of high river discharge ($Q_f$= 1800 m³/s).

The tidal effect is studied using three different tidal excursions. The tidal excursion values result in the plots that is shown in Figure 10. The longitudinal salinity distribution at tidal average conditions is independent of the tidal excursion as can be seen in Figure 10a. From Eq. (23), since the salinity amplitude coefficient $I_s$ is in direct proportion to the tidal excursion, the amplitude of the salinity shows a linearly increasing trend with the increased tidal excursion (Figure 10b).

## 5. Discussion

### 5.1. Time lag between salinity extremes and slack water

In estuaries, it is noticed that the maximum salinity appears after HWS and the minimum salinity appears before LWS. However, often, the salinity at HWS and LWS correspond approximately to the maximum and minimum salinity, respectively. The accuracy of this approximation cannot be inferred from existing steady-state models for salt intrusion, as time variation is neglected. As shown in Figure 11, the unsteady analytical solution proposed in this paper demonstrates that the phase lag between tidal velocity and salinity transportation is $\pi/2$, which means that the extreme values of salinity appear when the

tidal velocity is zero. Our unsteady equation for salinity (i.e. Eq. 11) demonstrates the influence of the river discharge on the occurrence of maximum and minimum salinity relative to HWS and LWS, respectively.

More generally, Eq. (14) offers a simple expression yielding qualitative insight into the role of the river discharge in the spatiotemporal variation of salinity in a well-mixed estuary. The time lag between salinity extremes and slack water is determined by the strength of the river flow, in a way that is consistent with the previous observations that the maximum salinity appears after HWS and the minimum salinity appears before LWS. The estimated river flow velocity at Huangpuyou station is about 1/6 of the tidal flow amplitude, resulting in a time lag between HW (at maximum salinity) and HWS (when total velocity is zero) of less than 30 minutes. At this station, it is acceptable to assume that the salinity reaches the maximum value at HWS and the minimum value at LWS.

## 5.2. Optimizing water intake

Estuaries are crucial feeding and breeding grounds for many life forms, and are a source of drinking water. Intrusion of salt water can temporarily halt the production of drinking water, and put stress on plant and animal species that have adapted to the typical salt concentrations along the estuary. In China, a value of 0.5 ‰ of salinity is considered as the upper limit of drinking water (SWEQ PRC, 2002), while turbot farmed in man-made ponds need to live in the water with less than 12 ‰ of salinity. The unsteady solution proposed in this paper shows to reproduce the intratidal variation of salt intrusion, which allows to estimate the window of opportunity for drinking water intake, and has the potential of application in aquaculture and water fetching works in estuaries.

Due to the serious increase of salt intrusion in recent years, the water intake from Humen estuary is more suitable for saline-water aquaculture rather than the residential use. However, the salinity along the estuarine channel is changing all the time according to the variations of the tides as well as the freshwater discharge. This makes it important to capture the temporal variation of salinity for optimizing the water intake of the man-made ponds around the estuary. The analytical model proposed in this study provides a simple and efficient approach to predicate the variation of salinity, which is economical and practical, with the limited amount of data available.

Close to Dasheng station, there is an aquaculture area with many man-made ponds of different sizes. Optimizing water intake is a key issue here. The applicatibity of the analytical model is illustrated by focussing on turbot farming ,which requires salinity of no less than 12‰. The observed salinity data on 29 January is used to calibrate the model, where the determination of three parameters is needed, i.e. tide-averaged salinity at mouth $\bar{s}_0$, the slope $k = \left( Q_f a / D A_0 \right)$ and the tidal excursion $E$. The decreasing trend of subtidal salinity is close to a linear relation from 29 January to 3 February (Figure 12b). Thus the tide-averaged salinity value of the predicted model is set as 90% of that on 29 January, considering the slight change of the subtidal salinity in the five days after 29 January. Moreover, the slope as well as the tidal excursion are assumed to be constant during the whole period from 29 January to 3 February. As shown in Figure 12c, the prediction by the model is in good agreement with the observation in this case. Furthermore, if more observed data is available to calibrate the tide-averaged salinity covering

the period from 29 January to 3 February, Eq. (14) performs better, as shown in Figure 12d. The available time for water intake can be obtained from the prediction, when the salinity concentration reaches a value higher than 12 ‰.

Since the fresh water discharge influences the slope (Brockway et al., 2012), it is reasonable to assume that the slope remains constant in a short time scale since the fresh water discharge variation has a time scale of days to months (Figure 12a). The tidal excursion is the integral over time of the tidal velocity between the low water slack and high water slack. It varies from day to day as the tidal wave changes from neap tide to spring tide (Savenije, 2005). Therefore, the tidal excursion is assumed to be independent of time in the neap cycle from 29 January to 3 February. Besides, Equation (18) is demonstrated to be a useful equation for the calculation of the tidal excursion, which offers an approach to estimate the tidal excursion with salinity data. The predicted salinity fits well with observed values, indicating that the estimation of the tide-averaged salinity during the neap tide is acceptable. However, the prediction accuracy of the model can be higher if more observed tide-averaged salinity data is available.

## 6. Conclusions

An unsteady-state analytical solution of salt intrusion is proposed based on the one-dimensional advection-diffusion equation for salinity, assuming a harmonic tidal wave with a single-frequency and a constant mixing coefficient. The predictive skill of the model has been illustrated from an application to the Humen Estuary, which shows it can offer an efficient approach to calculating the variation of salinity in a well-mixed estuary where the channel area is convergent. The results show that the analytical model is able to reproduce the intratidal variation of salt intrusion, and can be a useful tool to compute the time windows in which salinity remains below a critical threshold in an estuary.

**Author contributions**

YX and WZ formulated the overarching research goals and aims. YX, AH and WZ contributed to the development of the methodology. YX, AH, JZ and WZ discussed and interpreted the results. YX created the figures and wrote the original draft. AH, JZ, KK and WZ reviewed and edited the draft.

**Competing interests**

The authors declare no competing interests.

**Acknowledgement**

This work was supported by the "National Key R&D Program of China" [grant numbers 2017YFC0405900], the "Fundamental Research Funds for the Central Universities" [grant numbers 2018B56214, 2017B21514, 2018B13114], the "National Natural

Science Foundation of China" [grant numbers 41676078, 41506100], the "Open Foundation of Key Laboratory of Coastal Disasters and Defense of Ministry of Education" [grant numbers 201704] the "China Postdoctoral Science Foundation" [grant numbers 2017M621611], the "Open Research Foundation of Key Laboratory of the Pearl River Estuarine Dynamics and Associated Process Regulation, Ministry of Water Resources" [grant numbers 2018KJ05, 2017KJ04], and the "Project of Jiangsu Provincial Six Talent Peaks" [grant numbers XXRJ-008]. We thank the editor, Prof. Savenije, Dr. Cai and an anonymous reviewer for constructive comments on the initial draft of this manuscript.

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

**TABLES**

**Table 1** General information of hydrological stations in the Humen waterway

| Station name | Distance from the estuary mouth (km) | x-coordinate(m) [*] | y-coordinate(m) [*] |
|---|---|---|---|
| Dahu | 0 | 2524802 | 38459960 |
| Sishengwei | 9.9 | 2534512 | 38458163 |
| Zhangpeng | 18.4 | 2542539 | 38455607 |
| Machong | 25.4 | 2548948 | 38452466 |
| Dasheng | 28.0 | 2551430 | 38451984 |
| Huangpuyou | 36.9 | 2553758 | 38443358 |

[*] The coordinate system's origin is set at 22°05'12.9894"N, 113°27'34.9899"E.

**Table 2** Dispersion coefficient of salt intrusion in Humen estuary

| Data | River discharge $Q_f$ (m$^3$/s) | Tide range $H$ (m) | Slope $k$ | $R^2$ | Dispersion coefficient $D$ (m$^2$/s) |
|---|---|---|---|---|---|
| 29/01/2005 | 667 | 2.26 | -0.115 | 0.85 | 2562 |
| 30/01/2005 | 626 | 2.05 | -0.114 | 0.86 | 2425 |
| 31/01/2005 | 663 | 1.68 | -0.118 | 0.92 | 2481 |
| 01/02/2005 | 705 | 1.43 | -0.125 | 0.88 | 2492 |
| 02/02/2005 | 655 | 1.38 | -0.108 | 0.92 | 2678 |
| 03/02/2005 | 705 | 1.36 | -0.115 | 0.92 | 2708 |
| mean | | | | 0.89 | 2558 |

**Table 3** Calibrated values of Parameters

| Parameter | Unit | Value |
|---|---|---|
| $A_0$ | m$^2$ | 37822 |
| $a$ | km | 16.7 |
| $D$ | m$^2$/s | 2562 |
| $E_0$ | km | 26.7 |
| $e$ | km | 30 |
| $c$ | m/s | 12 |
| $\varphi_0$ | rad/s | -0.7 |

**Table 4** Calibration results of Song's model

| | Dahu | Sishengwei | Zhangpeng | Dasheng |
|---|---|---|---|---|
| $u_f$(m/s) | -0.0175 | -0.0317 | -0.0527 | -0.0937 |
| $\upsilon$ (m/s) | 1.8794 | 1.3512 | 1.0178 | 0.7390 |
| $D_S$(m$^2$/s) | 2269 | 2269 | 2269 | 2269 |

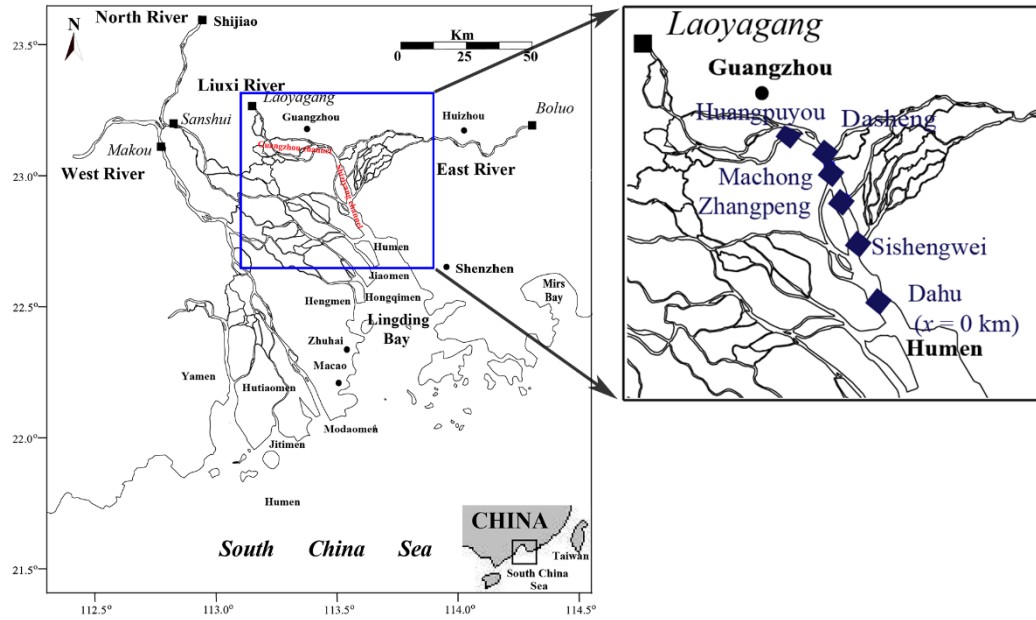

**Figure 1: Map of the Humen estuary, showing the gauging stations where salinity concentration was measured during the field survey from 29 January to 3 February, 2005.**

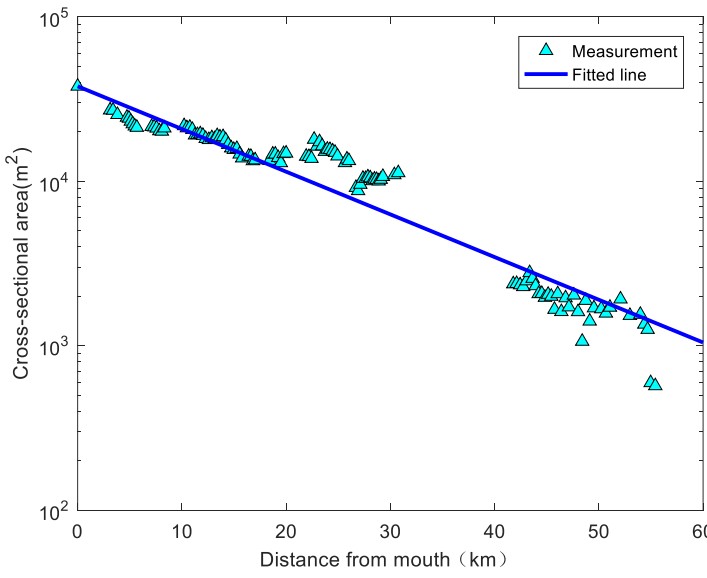

**Figure 2: Shape of the Humen estuary, showing the correlation between the cross-sectional area $A$ (m$^2$) and the distance from the estuary mouth $x$ (km). The coefficient of determination R$^2$ is 0.92. The triangles represent observations and the line represents the fit to Eq. (1), where the area at the estuary mouth $A_0$=37822 m$^2$ and the area convergence length ($a$) is 16.7 km.**

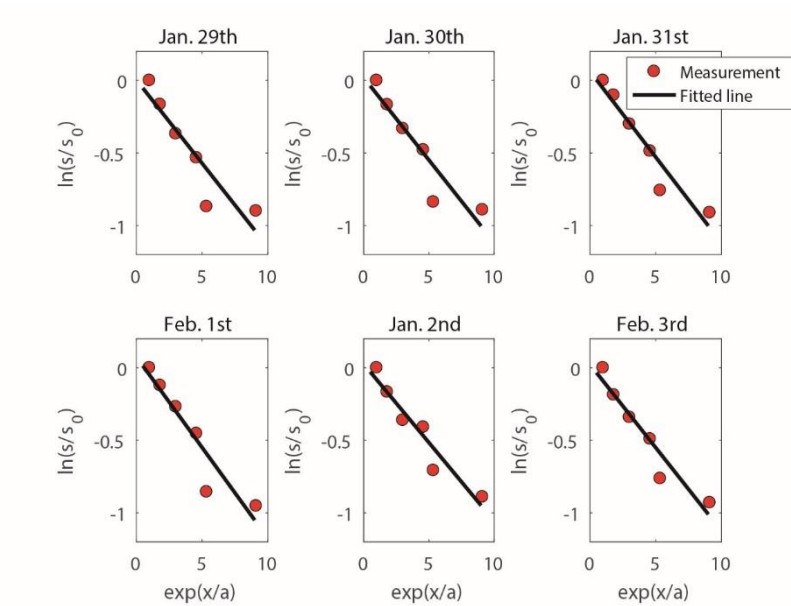

**Figure 3: Relative salinity concentration along the Humen estuary. The circlers represent observations and the lines represent the fit to Eq. (12).** $s$ **is the salinity at distance** $x$ **from the estuary mouth,** $s_0$ **is the salinity at the mouth and** $a$ **is the convergence length of the cross-section area.**

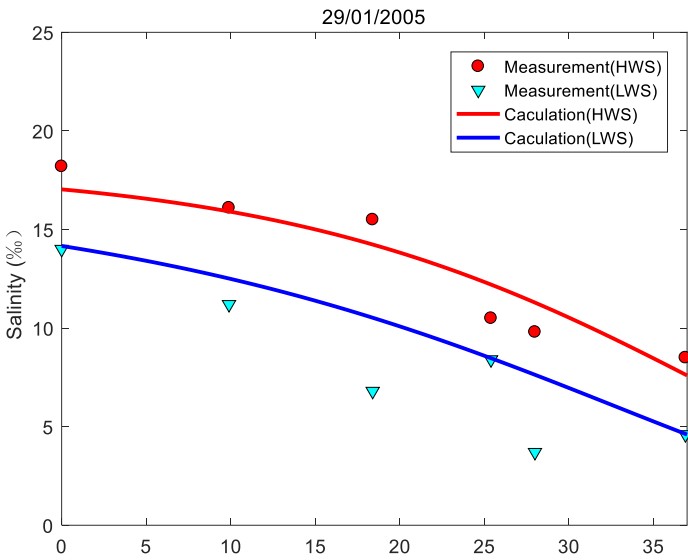

**Figure 4: Comparison between calibration results and measured salinity concentration along the river on 29 January, 2005, showing values of measured salinity at high water slack (circles) and low water slack (inverted triangles), and the calibrated salinity curves at high water slack (red curve) and low water slack (blue curve).**

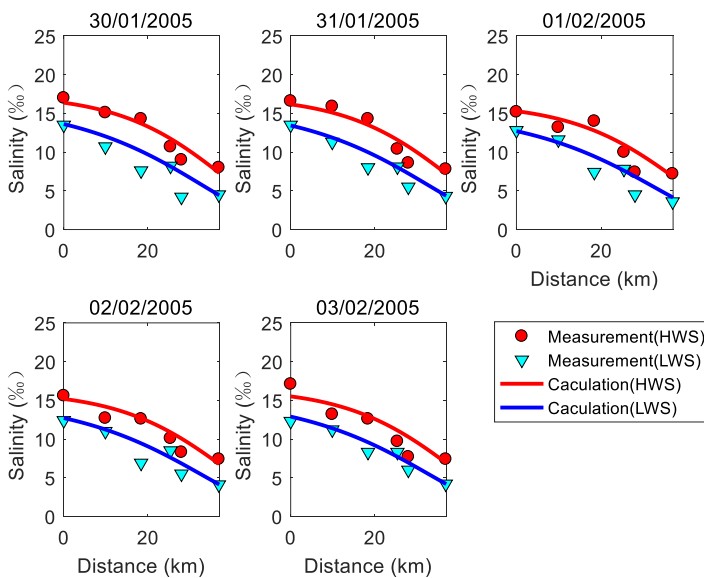

**Figure 5: Comparison between validation result and measured salinity concentration along the river from 30 January to 3 February, 2005.**

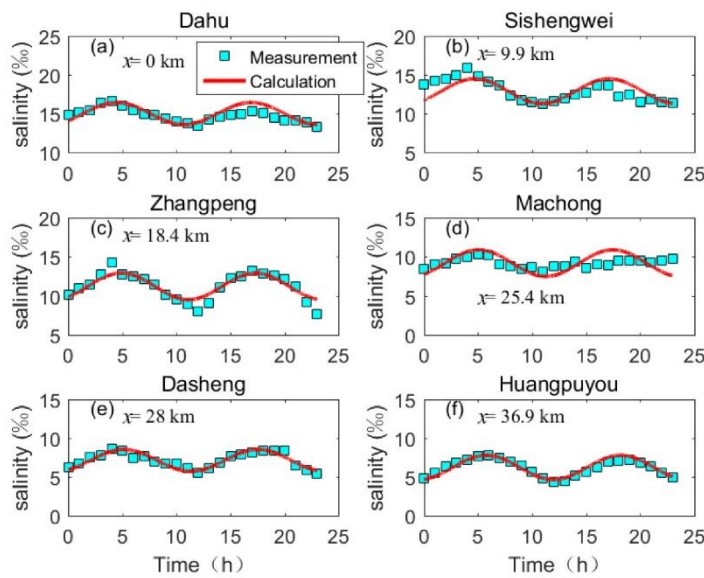

**Figure 6: Comparison between the predicted and measured salinity concentration over time on 31 January (neap tide) at each study site, showing that the analytical model captures the temporal variation of salinity. The hourly salinity measurements are represented by Rectangles, while the simulated salinity varying with time is represented by the red solid line. In the figure, *x* is the distance from the estuary mouth.**

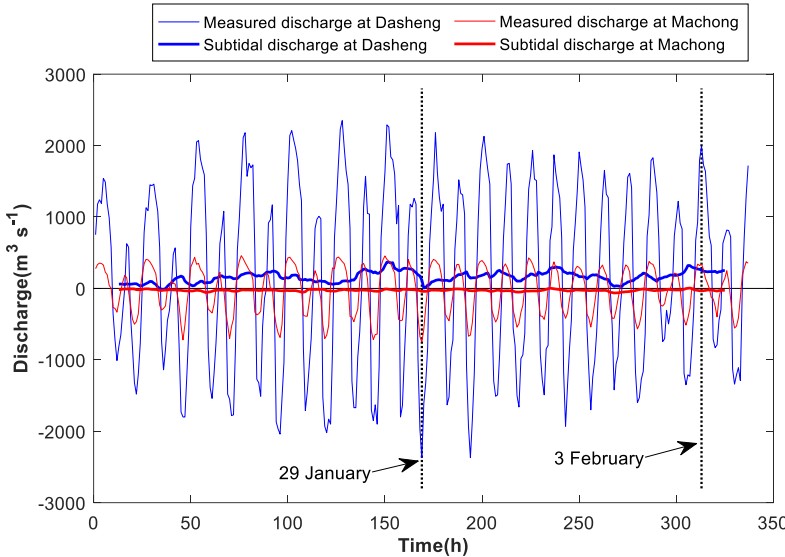

**Figure 7: Subtidal discharge measured at Machong station and Dasheng station from 29 January through 3 February. Positive values mean seaward.**

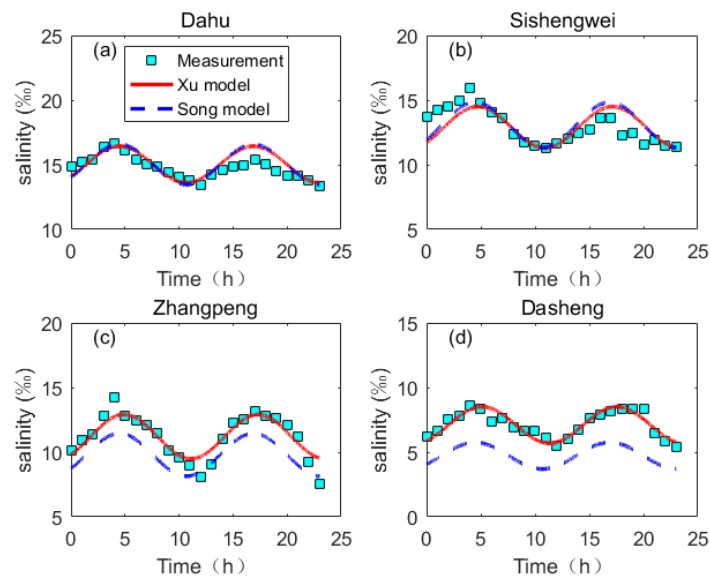

**Figure 8: Comparison between observed and computed salinity concentration over time on 31 January (neap tide) at study sites along the Humen estuary.**

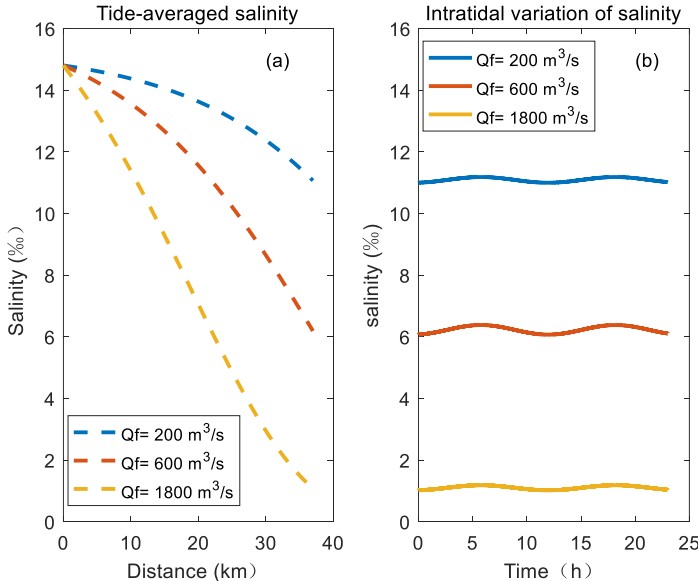

**Figure 9: (a) longitudinal salt intrusion curve at Tidal average considering different river discharge; (b) intratidal variation of salinity at Huangpuyou station on 31 January, 2005 considering different river discharge.**

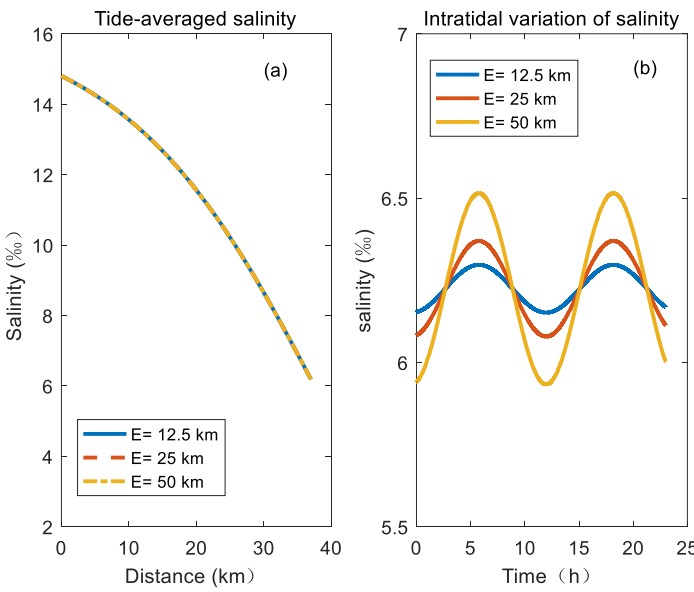

**Figure 10: (a) longitudinal salt intrusion curve at Tidal average considering different tidal excursion; (b) intratidal variation of salinity at Huangpuyou station on 31 January, 2005 considering different tidal excursion.**

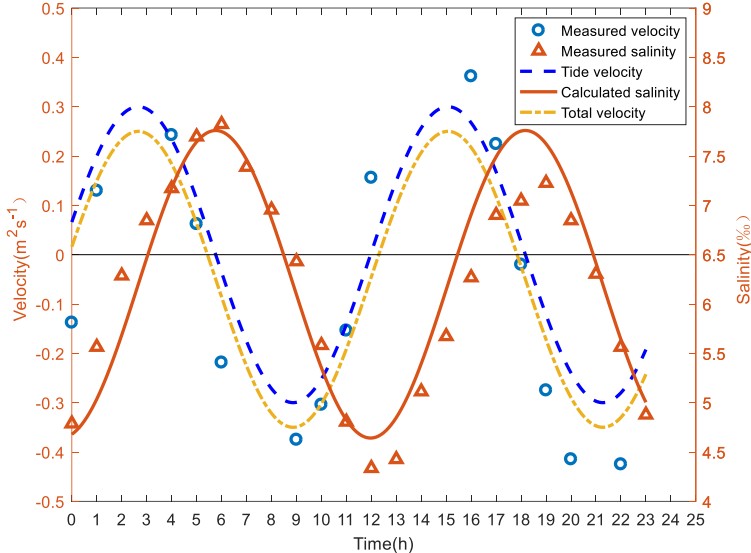

**Figure 11: Salinity and tidal flow velocity over a tidal cycle at Huangpuyou station. The measured salinity is represented by triangles and the measured flow velocity is indicated by circles (on 31 January 2005). The dashes line is the calculated tidal velocity while the dash-dotted line is the total velocity of tidal flow and river flow. The red solid curve represents salinity simulated by the unsteady analytical solution, which reproduces the time lag HWS and maximum salinity.**

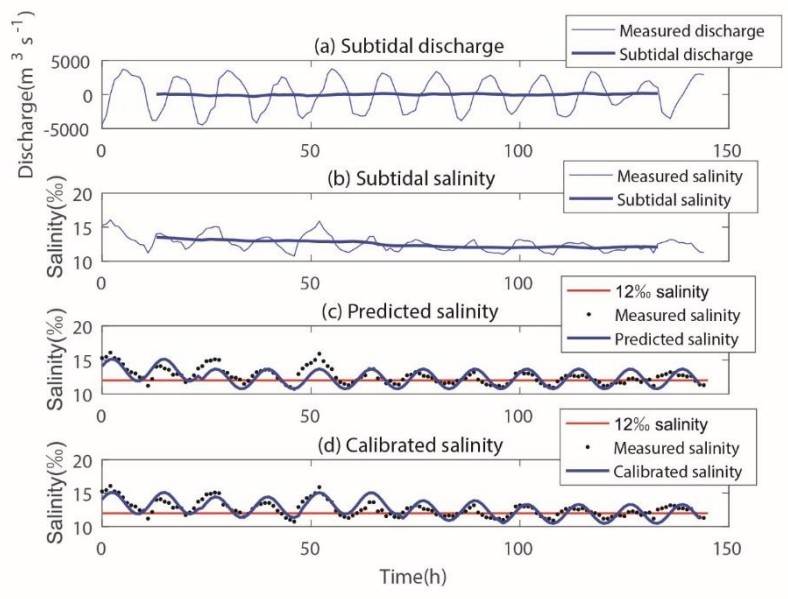

**Figure 12: Time for water intake of given salinity that is higher than 12‰. (a) Slight changes of the subtidal discharge; (b) Decreasing trend of subtidal salinity; (c) Predicted salinity on the basis of observed data on 29 January; (d) Calibrated salinity on the basis of observed data from 29 January to 3 February 2005.**