# Peer review of "Analytical model captures intratidal variation of salinity in a convergent, well-mixed estuary"

_Hydrology and Earth System Sciences, 2019_

## Referee Comment (RC1) · Huayang Cai (Referee) · 5 Jul 2019

**Review of "Analytical model captures intratidal variation of salinity in a convergent, well-mixed estuary" by Xu et al.**

In this paper, the authors proposed an unsteady analytical model for salt intrusion to understand the spatial-temporal dynamics of salt transport under different riverine and tidal forcing. The model was applied to the Humen estuary, which is a tide-dominated and well-mixed estuary. And the modelled results correspond well with the observed data. The paper is interesting and of important scientific implications for estuarine dynamics. However, there are still some major concerns that should be properly addressed before the paper can be accepted in this journal. Thus, I would suggest the authors to have a substantial revision.

Major concerns:

1.  Method in Section 2: It is noted that a rather similar approach for salinity intrusion in an unsteady state was proposed by Song et al. (2008) entitled "One-dimensional unsteady analytical solution of salinity intrusion in estuaries". It is better to clarify the main difference between the current model and the one proposed by Song et al. (2008) in order to highlight the new insights into the salt dynamics.

2.  P8, Lines 19-24, estimation of the tidal excursion: Note that the tidal excursion is a critical parameter that links the salinity intrusion to the tidal hydrodynamics. In this study, the authors assumed that the longitudinal tidal excursion can be described an exponential function. However, such an assumption is only reasonable for a short channel (Let's say less than 10 km). I would suggest the authors to adopt an analytical hydrodynamics model to reproduce the longitudinal tidal excursion since there exists a long traditional analytical solution for tidal hydrodynamics in estuaries (e.g., Toffolon and Savenije, 2011; Winterwerp and Wang, 2013; Cai et al., 2016). The advantage of coupling the salinity intrusion model to the tidal dynamics lies in that it enables directly linking the salt dynamics into the tidal forcing (e.g., tidal amplitude imposed at the estuary mouth). Moreover, it allows to have a prediction of salinity intrusion for different tidal forcing conditions (e.g., neap-spring changes) for given tidal amplitude observed at the estuary mouth. The current model used the observed salinity to forecast the tidal excursion (i.e., Eq. 18 in the manuscript), which is not very practical if prediction is required.

3.  P8, Lines 25-29, estimation of the wave celerity: Similar to the tidal excursion, I would suggest the authors to link the wave celerity to the tidal forcing imposed at the estuary mouth by means of an analytical model for tidal hydrodynamics.

4.  For the time being, the authors only illustrate the proposed analytical model applied to the Humen estuary during the neap tide condition, when the salt intrusion length is approximately minimum. I would suggest the authors to adopt the model to the case during the spring tide condition when the salt intrusion really matters. In section 4.2 concerning the model validation, since the authors only used the dataset from Jan. 29th to Feb. 3rd, I think this is only kind of the model calibration rather than validation because the tidal hydrodynamics is more or less the same during the chosen period.

5.  Sensitivity analysis: As mentioned by the authors, the proposed analytical model can directly reflect the influence of the tide and the interaction between the tide and runoff (see Abstract part

in Line 17). Hence, it is better to conduct a sensitivity analysis of the salinity distribution to both the tidal and riverine forcing imposed at both ends of the estuary.

Minor concerns:

1. P3, Eq. (3) and Eq. (4): Here please clarify the physical meaning of $s_1$ and $s_2$ coefficients. In addition, it is noted that the salinity and velocity are assumed to be in phase since they have the same initial phase, am I right? Please also clarify this important assumption.

2. P7, Line 18: Please clarify where the salinity was sampled. It was sampled in the central part of the channel or near side banks? Due to the fact that the model used the cross-sectional averaged salinity concentration, it would be better to clarify this point.

3. P9, Lines 14-16: It is better to illustrate the stratification or mixing during the studied period since the authors already collected both the surface and bottom salinity concentration.

4. P11, Lines 6-8: Due to the assumptions of Eqs. (3)-(4), the extreme values of salinity appear when the tidal velocity is zero.

5. Figure 1: Please use 'West River' and 'North River' instead of 'Xijiang River' and 'Beijiang River', respectively. Meanwhile, it is better to indicate the locations of outlets that were mentioned in the main text.

6. Figure 2: It is better to use the logarithm scale.

7. Figures 4, 6, 7: Please relocate the legend to a suitable place.

References:

Cai, H., Toffolon, M., Savenije, H.H.G., 2016a. An analytical approach to determining resonance in semi-closed convergent tidal channels, Coast Eng. J., 58(03), 1650009.

Toffolon, M., Savenije, H.H.G., 2011. Revisiting linearized one-dimensional tidal propagation. J. Geophys Res., 116. DOI:ArtnC0700710.1029/2010jc006616

Winterwerp, J.C., Wang, Z.B., 2013. Man-induced regime shifts in small estuaries-I: theory. Ocean Dynam., 63(11-12): 1279-1292. DOI:10.1007/s10236-013-0662-9

---

## Referee Comment (RC2) · Anonymous Referee #2 · 10 Jul 2019

**Review of "Analytical model captures intratidal variation of salinity in a convergent, well-mixed estuary" by Xu et al.**

In this manuscript, an unsteady analytical solution was presented to simulate the spatial-temporal variation of salinity in convergent estuaries and applied to the Humen estuary of the Pearl River Delta. There are a lot of issues which should be addressed.

**Major points:**

1. This manuscript is about the unsteady state analytical model for salt intrusion, but in the introduction section there is no anything about unsteady state analytical model. Nobody else did the unsteady state analytical model?

2. What differences are there between your model and previous models? What are the advantages of your model? Authors should compare your model results with other model results, to prove that your model is better.

3. In the methods section, which are input parameters, and how to determine them? These should be presented clearly.

4. In the application of the model to the Humen estuary, the first location of measurements (Dahu, figure 1) was set as the mouth of the estuary, and authors only calculated the results between station 1 and station 6 (figure 4). Actually, the real mouth is far downstream from station 1.

5. In figure 2, cross-sectional area of the Humen estuary was only shown for the reach between 0 km to 60 km. However, the Humen channel has a total length of 128 km (page 7, line 4). I think that the mouth in figure 2 should be the same as that in figure 4. If only part topography data was used, the area convergence length you obtained may be not correct. It is an important parameter in the model.

6. Section 4.1 (Application to the Humen Estuary) is about calibration of model. Authors only discussed the calibration of parameters. The calibration results of model were shown in section 4.2 (model validation). In other words, model calibration and model validation used the same data. Although in figure 4 the results between 29 January and 3 February were shown, the conditions were similar.

7. Section 6 Conclusions. In this manuscript, the main work is application of the model to the Humen estuary, showing calibration results. The first paragraph is

enough. In the second paragraph, part is about results instead of conclusions, and the other part is already in the first paragraph. In addition, "predictive", "predicating", and "predictable" used in conclusions are not proper.

**Minor points:**

1. Page 1, lines 16-17: "Compared with steady-state solutions, it can directly reflect the influence of the tide and the interaction between the tide and runoff". Salt intrusion is the result of interaction between tide and runoff. The steady-state solution cannot reflect the influence of tide and interaction of tide and runoff? And authors did not compare their solution with steady-state solutions.

2. Page 1, line 31 and page 2, line 1: "Hence, the effects of human activities on salt intrusion are of major interest to engineers and scientists". This sentence is not related to the topic of this manuscript. Authors did not do anything about the influence of human activities.

3. Page 3, lines 2-5. The sentences about paper organization are not necessary.

4. Page 6. What is *e* in equations 17 and 18?

5. Page 6, line 20. Here the citation of a reference is not necessary. Particularly the reference is from a foreigner. Is a foreigner more familiar with a Chinese estuary?

6. Page 7, line 6. What does the ES mean?

7. Page 7, lines 16-17: "The Humen waterway is well-mixed in the dry seasons (Luo et al., 2010)". The mixing condition can be seen directly from the vertical distribution of salinity, which should be shown in section 3.1 (overview of the study area).

8. Page 7, section 3.2 data. What data about the tide was used in this study? In line 12, it is tidal flow. But in line 19, it is tidal level.

9. The title of section 4.1 can be changed into "Model calibration", corresponding with section 4.2 Model validation.

10. Page 9, lines 18-19. Why did you use the daily maximum and minimum salinity in figure 4?

11. Page 9, the last paragraph. I did not understand what authors wanted to express except for the first sentence. In the first sentence, the "downstream" is relative to the 40 km reach in figure 4 or the whole channel? It can be seen from figure 4 that the main overestimates occur at station 3 and station 5.

12. Page 10, line 16: "salinity variation is more symmetrical further away from the study site". What does this sentence mean? It is difficult to understand.

13. Page 10, lines 29-30. Authors used this sentence to explain the nonperiodic variation of salinity at Machong station in figure 5. It seems that only in the second tidal cycle, the variation is abnormal.

14. Page 16, table 2. All parameters used in the model should be shown.

15. Page 17, figure 1.

    (1) The Pearl River estuary is too complicated, and Humen is only one of eight branches. The figure caption is map of Humen estuary. But where is Humen? Only six gauging stations can be seen. The Humen estuary should be enlarged and shown clearly.

    (2) River names "Beijiang River and Xijiang River" are different from the names "the North river and West river" in the text.

    (3) East River and the Shiziyang channel in Page 10, line 23 were not shown in figure 1.

16. Page 19, caption of figure 3: "The linear relationship between these quantities predicted by Eq. (12) has been confirmed for all surveys, and the figures here show the linear line fitting results from Jan. 29th to Feb. 3$^{rd}$". Page 22, caption of figure 6: "The subtidal discharge switches from seaward to landward between Machong and Dasheng stations, which will have an impact on salinity dynamics." These sentences should not be in the figure caption.

17. The legends should be inside or outside figures, instead of covering the curves or words, such as figure 4, figure 6, and figure 7.

18. Is Humen a waterway or estuary? In some figure and table captions waterway was used, but in others estuary was used. It is the same in the text.

19. English writing should be improved. For examples:

    (1) Page 7, line 19, "salinity was obtained by using a salimeter". "by" or "using" is enough.

    (2) Page 9 and page 10. "Analysis of " in the titles of section 4.2.1 and 4.2.2 can be deleted. They are not necessary.

    (3) Page 12, line 8, "the predicted result obtained by this model". "predicted" or "obtained" is enough.

    (4) Page 16, caption of table 2: "Values of the parameters of salt intrusion in Humen estuary". "Values of the" can be deleted, "parameters" is enough.

    These are only examples. Authors should check every sentence to make them standard, concise, and fluency.

---

## Author Comment (AC1) · 3 Sep 2019

**Reviewer # 1 Questions and our responses**

We thank Reviewer #1 for constructive comments and suggestions to improve our paper. In this section, we first list reviewer's questions/comments, and then provide our answers. The questions/comments are in italics, and our responses are in bold text.

*In this paper, the authors proposed an unsteady analytical model for salt intrusion to understand the spatial-temporal dynamics of salt transport under different riverine and tidal forcing. The model was applied to the Humen estuary, which is a tide-dominated and well-mixed estuary. And the modelled results correspond well with the observed data. The paper is interesting and of important scientific implications for estuarine dynamics. However, there are still some major concerns that should be properly addressed before the paper can be accepted in this journal. Thus, I would suggest the authors to have a substantial revision.*

*Major concerns:*
*1. Method in Section 2: It is noted that a rather similar approach for salinity intrusion in an unsteady state was proposed by Song et al. (2008) entitled "One-dimensional unsteady analytical solution of salinity intrusion in estuaries". It is better to clarify the main difference between the current model and the one proposed by Song et al. (2008) in order to highlight the new insights into the salt dynamics.*

**The unsteady analytical model developed by Song et al. (2008) can reproduce the salinity process in an idealized estuary with constant depth and constant width. Song's model is thus applicable to laboratory flumes and artificial channels. However, the channel cross-sectional area of estuaries is typically converging. One innovation of our paper is to better capture the natural topography of alluvial estuaries, assuming the cross-sectional area to obey an exponential function. So, our paper continues on Song's work within the geometrical setting of an alluvial estuary. We will clarify this in the revision.**

**Song, Z. Y., Huang, X. J., Zhang, H. G., Chen, X. Q., and Kong, J.: One–dimensional unsteady analytical solution of salinity intrusion in estuaries, China Ocean Eng., 22, 113–122, 2008.**

*2. P8, Lines 19-24, estimation of the tidal excursion: Note that the tidal excursion is a critical parameter that links the salinity intrusion to the tidal hydrodynamics. In this study, the authors assumed that the longitudinal tidal excursion can be described an exponential function. However, such an assumption is only reasonable for a short channel (Let's say less than 10 km). I would suggest the authors to adopt an analytical hydrodynamics model to reproduce the longitudinal tidal excursion since there exists a long traditional analytical solution for tidal hydrodynamics in estuaries (e.g., Toffolon and Savenije, 2011; Winterwerp and Wang, 2013; Cai et al., 2016). The advantage of coupling the salinity intrusion model to the tidal dynamics lies in that it enables directly linking the salt dynamics into the tidal forcing (e.g., tidal amplitude imposed at the estuary mouth). Moreover, it allows to have a prediction of salinity intrusion for*

*different tidal forcing conditions (e.g., neap-spring changes) for given tidal amplitude observed at the estuary mouth. The current model used the observed salinity to forecast the tidal excursion (i.e., Eq. 18 in the manuscript), which is not very practical if prediction is required.*

**We appreciate the reviewer's suggestion and will use an analytical model for tidal hydrodynamics to compute the excursion length in the revised version. In the Method section, we will introduce Cai's method (Cai et al., 2016) as below:**

**"Tidal wave propagation can be described analytically by a set of four implicit equations (Cai et al., 2012), the phase lag equation $\tan(\varepsilon) = \lambda/(\gamma - \delta)$, the scaling equation $\mu = \sin(\varepsilon)/\lambda$, the damping equation $\delta = \gamma/2 - 4\chi\mu/(9\pi\lambda) - \chi\mu^2/3$, and the celerity equation $\lambda^2 = 1 - \delta(\lambda - \delta)$, where $\lambda$ is the celerity number $\lambda = c_0/c$, $\mu$ is the velocity number $\mu = \upsilon\bar{h}/(r_s\eta c_0)$, $\delta$ is the damping number $\delta = c_0 \mathrm{d}\eta/(\eta \mathrm{d}x\omega)$, and $\varepsilon$ is the phase lag between HW and HWS $\varepsilon = \pi/2 - (\phi_Z - \phi_U)$. Here, three dimensionless parameters control the tidal hydrodynamics (Savenije et al., 2008), i.e. the dimensionless tidal amplitude $\zeta = \eta/\bar{h}$, the estuary shape number $\gamma = c_0/(\omega a)$ and the friction number $\chi = r_s g c_0 \zeta / \left( K_s^2 \omega \bar{h}^{4/3} \right) \left[ 1 - (4\zeta/3)^2 \right]^{-1}$, where $\eta$ is the tidal amplitude, $K_S$ is the Manning-Strickler friction coefficient, $r_s$ the storage width ratio, $\bar{h}$ is the tide-averaged depth and $c_0$ is the classical wave celerity $c_0 = \sqrt{g\bar{h}/r_s}$. Then, with the available geometry and friction data at the estuary mouth, the tidal propagation celerity and the tidal amplitude (or the tidal excursion) can be obtained by solving the set of four equations."**

**In addition, we will apply the hydrodynamics model proposed by Cai et al. (2016) to the Humen estuary based on observations. The averaged depth along the axis of the Humen estuary is shown in Figure S1. The coefficient of determination $R^2$ is 0.67, which indicates that the topography is too complex to be represented by an exponential function. The water depth is influenced greatly by human activities. The input parameters used for the tidal hydrodynamics model will be summarized in Table S1. A new Figure S2a shows the computed tidal amplitude and the tidal excursion obtained with a hybrid model, using a variable depth along the estuary (Cai et al., 2012). The tidal amplitude along the Humen estuary can be well simulated by the analytical model while the tidal excursion is underestimated. There are two reasons which may cause the underestimation. The first one is due to the inaccurate estimation of the averaged depth along the estuary, since the depth convergence length $d$ cannot be fitted well to the exponential function $\bar{h} = \bar{h}_0 \exp(x/-d)$ in the Humen estuary. It can have a serious impact on the three dimensionless parameters which control the tidal hydrodynamics, i.e. the dimensionless tidal amplitude, the estuary shape number and the friction number. The other one is the assumption that the tidal excursion is independent of the distance along the Humen estuary. Well, as was shown by Savenije (2005), the tidal excursion can be assumed to be constant in many estuaries worldwide. However, a conclusion from the measurement data show that the tidal excursion may be damped along the estuary, like in the Mekong Estuary. In four estuary branches, over more than 100 kilometers, tidal excursion decay can be described by an**

exponential function (Nguyen, 2008). In Figure S3a we will present the computed salinity curves using the calibrated variable tidal excursion along the Humen estuary. It better reproduces the salinity at HWS and LWS compared with the results obtained based on a hybrid model shown in Figure S3b. Therefore, similarly, the observations in the Humen estuary indicate that the tidal excursion decreases exponentially.

We agree with the reviewer that it is important to link the salinity intrusion to the tidal hydrodynamics. It can help the model to become applicable to a wider range of flow conditions. Therefore, the analytical hydrodynamics model by Cai et al. (2012) will be presented in the revised methods section. We will offer an analytical approach to reproduce the main tidal dynamics coupling to the salt dynamics. However, unfortunately, the analytical model for tidal dynamics cannot be used in this case, due to the limitations of available geometry as well as the assumption of the tidal excursion. So, in Section 4, we will provide another way to calibrate the tidal excursion based on the measurements of salinity.

Cai, H., Savenije, H. H. G., and Toffolon, M.: A new analytical framework for assessing the effect of sea-level rise and dredging on tidal damping in estuaries, J. Geophys. Res., 117, C09023, doi:10.1029/2012JC008000, 2012.
Cai, H., Toffolon, M., and Savenije, H.H.G.: An analytical approach to determining resonance in semi-closed convergent tidal channels, Coast Eng. J., 58(03), 1650009, 2016.
Nguyen, A. D.: Salt Intrusion, Tides and Mixing in Multi-Channel Estuaries: PhD: UNESCO-IHE Institute, Delft. CRC Press, 2008, p52.
Savenije, H. H. G.: Salinity and Tides in Alluvial Estuaries, Elsevier, Amsterdam, 2005.

[Figure]

**Figure S1: Shape of the Humen estuary, showing the correlation between the cross-sectional area $A$ (m²) and averaged depth $\bar{h}$ along the estuary axis with fitted trend lines.**

[Figure]

**Figure S2: (a) The computed tidal amplitude and tidal excursion in the Humen estuary based on a hybrid model using a variable depth along the estuary; (b) Comparison between computed salinity and the observations at HWS and LWS. The tidal excursion is computed with a hybrid model.**

[Figure]

**Figure S3: (a) Comparison between the observed and computed salinity curve using the calibrated tidal excursion; (b) Comparison between the observed and computed salinity curve based on a hybrid model using a variable depth along the estuary.**

**Table S1: Inputs used for the tidal hydrodynamics model**

|  | $A$ | $a$ | $h_0$ | $d$ | $\eta_0$ | $Ks$ | $T$ | $s_0$ |
|--|-----|-----|-------|-----|----------|------|-----|-------|
|  | $(m^2)$ | (km) | (m) | (km) | (m) | $(m^{1/3}s^{-1})$ | (s) | (‰) |
| **Humen** | 37822 | 16.7 | 10 | 50 | 0.84 | 35 | 44400 | 15.02 |

*3. P8, Lines 25-29, estimation of the wave celerity: Similar to the tidal excursion, I would suggest the authors to link the wave celerity to the tidal forcing imposed at the estuary mouth by means of an analytical model for tidal hydrodynamics.*

**The analytical hydrodynamics model (Cai et al., 2012) will be presented in the method section in the revised version, but it does not apply to our specific field case.**

**Cai, H., Savenije, H. H. G., and Toffolon, M.: A new analytical framework for assessing the effect of sea-level rise and dredging on tidal damping in estuaries, J. Geophys. Res., 117, C09023, doi:10.1029/2012JC008000, 2012.**

*4. For the time being, the authors only illustrate the proposed analytical model applied to the Humen estuary during the neap tide condition, when the salt intrusion length is approximately minimum. I would suggest the authors to adopt the model to the case during the spring tide condition when the salt intrusion really matters. In section 4.2 concerning the model validation, since the authors only used the dataset from Jan. 29th to Feb. 3rd, I think this is only kind of the model calibration rather than validation because the tidal hydrodynamics is more or less the same during the chosen period.*

**In the study, the salinity is assumed to be forced by a harmonic tidal wave with a single-frequency. So our model is more applicable for estuaries with semidiurnal tides or diurnal tides. It is found that the semidiurnal tide is the distinctively dominant tidal wave in the neap tide in Humen, while the diurnal tide is as important as the semidiurnal tide in the spring tide. Therefore, we chose the data during the neap tide to illustrate our unsteady model in this case.**

**We have rewritten Section 4 to make it clearer in the revised version. The calibrated parameters include tidal excursion *E* and dispersion coefficient *D*. Although each of the calibrated dispersion coefficients from 29 January to 3 February was listed in Table 2, in fact, only the one on 29 January was used for calibration. In other words, we use the data on 29 January to calibrate the parameters of the model and use the data from 30 January to 3 February to validate the model. To make it clearer, in the revised version, we use two figures to show the results, Figure 4 is the calibrated result and Figure 5 is the validation results, as below:**

[Figure]

**Figure 4: Comparison between calibration result and measured salinity concentration along the river on 29 January, 2005, showing values of measured salinity at high water slack (circle) and low water slack (inverted triangle), and the calibrated salinity curves at high water slack (red curve) and low water slack (blue curve).**

[Figure]

**Figure 5: Comparison between validation result and measured salinity concentration along the river from 30 January to 3 February, 2005.**

*5. Sensitivity analysis: As mentioned by the authors, the proposed analytical model can directly reflect the influence of the tide and the interaction between the tide and runoff (see Abstract part in Line 17). Hence, it is better to conduct a sensitivity analysis of the salinity distribution to both the tidal and riverine forcing imposed at both ends of the estuary.*

**We appreciate the reviewer's suggestion and add a "Sensitivity analysis" part in the revised version, as below:**

"The amplitude of salinity can be described by:

$$\hat{s} = \overline{s}_x * I_s, \tag{23}$$

where $\overline{s}_x$ is the tide-averaged salinity along the estuary and is a function of the river discharge, i.e. Eq.(12). The parameter $I_s$ is the salinity amplitude coefficient that is defined as:

$$I_s = -\frac{EQ_f}{2DA}, \tag{24}$$

representing the interaction between the tides and the river discharge. To investigate the longitudinal salinity distribution and intratidal salinity variation for different discharge and tidal dynamic conditions in the Humen estuary, Eqs. (12) and (23) are used to plot the longitudinal salinity curve and intratidal variation of salinity, respectively. The implemented parameters are the same as shown in Table 3, only the river discharge and the tidal excursion are variable.

Three constant discharge values of 200, 600 and 1800 $m^3$/s are used to evaluate the impact of the river discharge on the salinity variation. The discharges are chosen because the minimum discharge in the dry season is around 600 $m^3$/s in the Humen estuary, and low salinity can be measured at Huangpuyou station when the discharge is larger than 1800 $m^3$/s. In addition, the discharge in the extreme dry season is set to be 200 $m^3$/s. The longitudinal salinity curve can be seen Figure 9a. At tidal average conditions, the salt intrusion length gets smaller when the discharge increases. The steepest salinity gradient can be found at the highest discharge ($Q_f$=1800 $m^3$/s). It is clear from Figure 9b that the salinity amplitude increases firstly and then decreases as the river discharge increases. This is because during periods of low river discharge ($Q_f$= 200 $m^3$/s), the tide-averaged salinity is larger but the salinity amplitude coefficient $I_s$ is smaller, which indicates the weaker interaction between the river flow and the tides. However, the tide-averaged salinity decreases rapidly with the increasing river discharge as we can see from Figure 9a, resulting in a smaller amplitude of salinity during periods of high river discharge ($Q_f$= 1800 $m^3$/s).

The tidal effect is studied using three different tidal excursions. The tidal excursion values result in the plots that is shown in Figure 10. The longitudinal salinity distribution at tidal average conditions is independent of the tidal excursion, as can be seen in Figure 10a.   From Eq. (23), since the salinity amplitude coefficient $I_s$ is in direct proportion to the tidal excursion, the amplitude of the salinity shows a linearly increasing trend with the increased tidal excursion (Figure 10b)."

[Figure]

**Figure 9: (a) Longitudinal salt intrusion curve at Tidal average considering different river discharge; (b) intratidal variation of salinity at Huangpuyou station on 31 January, 2005 considering different river discharge.**

[Figure]

**Figure 10: (a) Longitudinal salt intrusion curve at Tidal average considering different tidal excursion; (b) intratidal variation of salinity at Huangpuyou station on 31 January, 2005 considering different tidal excursion.**

*Minor concerns:*
*1. P3, Eq. (3) and Eq. (4): Here please clarify the physical meaning of s1 and s2 coefficients. In addition, it is noted that the salinity and velocity are assumed to be in phase since they have the same initial phase, am I right? Please also clarify this important assumption.*

**In fact, Eq. (3) is the expression of salinity using a first-order Fourier expansion method. Therefore, mathematically, s1 and s2 are the Fourier**

expanding coefficients, and the physical meaning is the amplitude of salinity variation. Since the value of the initial phase in Eq. (3) has no impact on the Fourier expansion, we assume the salinity having the same initial phase with the velocity for convenience of calculating.

*2. P7, Line 18: Please clarify where the salinity was sampled. It was sampled in the central part of the channel or near side banks? Due to the fact that the model used the cross-sectional averaged salinity concentration, it would be better to clarify this point.*

Considering the impact of the shipping, the measuring positions were near the banks, with certain distances ranging from 605 m to 70 m.

*3. P9, Lines 14-16: It is better to illustrate the stratification or mixing during the studied period since the authors already collected both the surface and bottom salinity concentration.*

The field survey was carried out by Guangdong Province Hydrology Bureau and the Pearl Hydrology Bureau from the River Conservancy Commission. Unfortunately, they only provided us the vertically averaged salinity at each measuring location, related to the well-mixed condition in Humen estuary. For lack of the vertical salinity data, we further support this view in the revised version, as below:

"…The Humen estuary is well-mixed under normal flow conditions during the dry season (Ou, 2009; Luo et al., 2010). Due to three year's drought, the river discharge decreased by 30 to 50 percent during the study period in 2005 compared to a normal year (Liao, Pan, and Dong, 2008). Thus, it was well mixed during the calibration and validation…"

Liao, D.Y.; Pan, T.J., and Dong, Y.L., 2008. Characteristics of salt intrusion and its impact analysis in Guangzhou. Environment, S1, 4-5. (In Chinese)

Luo, L., Chen, J., Yang, W., and Wang, D.X, 2010. An intensive saltwater intrusion in the pearl river delta during the winter of 2007–2008, J. Trop. Oceanogr., 6, 22-28. (In Chinese)

Ou, S.Y., 2009. Spatial difference about activity of saline water intrusion in the Pearl River Delta. Scientia Geographica Sinica, 29(1), 89-92. (In Chinese)

*4. P11, Lines 6-8: Due to the assumptions of Eqs. (3)-(4), the extreme values of salinity appear when the tidal velocity is zero.*

In physical terms, the tidal flow moves in the reversed direction just in the next tick when the tidal velocity turns into zero. At that moment, the salinity at the study site is the maximum or minimum value. Savenije (2005) also assumed that the maximum salinity is reached when tidal discharge is zero.

Savenije, H. H. G.: Salinity and Tides in Alluvial Estuaries, Elsevier, Amsterdam, 2005, p141.

*5. Figure 1: Please use 'West River' and 'North River' instead of 'Xijiang River' and 'Beijiang River', respectively. Meanwhile, it is better to indicate the locations of outlets*

*that were mentioned in the main text.*

We appreciate the reviewer's suggestion and redraw Figure 1 in the revised version as below:

[Figure]

**Figure 1: Map of the Humen estuary, showing the gauging stations where salinity concentration was measured during the field survey from 29 January to 3 February, 2005.**

*6. Figure 2: It is better to use the logarithm scale.*

We will use a logarithmic scale in the revised version as below:

[Figure]

**Figure 2: Shape of the Humen estuary, showing the correlation between the cross-sectional area $A$ (m$^2$) and the distance from the estuary mouth $x$ (km). The coefficient of determination $R^2$ is 0.92. The triangles represent observations and the line represents the fit to Eq. (1), where the area at the estuary mouth $A_0$=37822 m$^2$ and the area convergence length ($a$) is 16.7 km.**

*7. Figures 4, 6, 7: Please relocate the legend to a suitable place.*

We appreciate the reviewer's suggestion and redraw Figures 4, 6 and 7 in the

**revised version as below:**

[Figure]

**Figure 4: Comparison between calibration results and measured salinity concentration along the river on 29 January, 2005, showing values of measured salinity at high water slack (circles) and low water slack (inverted triangles), and the calibrated salinity curves at high water slack (red curve) and low water slack (blue curve).**

[Figure]

**Figure 5: Comparison between validation result and measured salinity concentration along the river from 30 January to 3 February, 2005.**

[Figure]

**Figure 11: Subtidal discharge measured at Machong station and Dasheng station from 29 January through 3 February. Positive values mean seaward.**

[Figure]

**Figure 12: Salinity and tidal flow velocity over a tidal cycle at Huangpuyou station. The measured salinity is represented by triangles and the measured flow velocity is indicated by circles (on 31 January 2005). The dashes line is the calculated tidal velocity while the dash-dotted line is the total velocity of tidal flow and river flow. The red solid curve represents salinity simulated by the unsteady analytical solution, which reproduces the time lag HWS and maximum salinity.**

*References::*

*Cai, H., Toffolon, M., Savenije, H.H.G., 2016a. An analytical approach to determining resonance in semi-closed convergent tidal channels, Coast Eng. J., 58(03), 1650009.*

*Toffolon, M., Savenije, H.H.G., 2011. Revisiting linearized one-dimensional tidal propagation. J. Geophys Res., 116. DOI:ArtnC0700710.1029/2010jc006616*

*Winterwerp, J.C., Wang, Z.B., 2013. Man-induced regime shifts in small estuaries-I: theory. Ocean Dynam., 63(11-12): 1279-1292. DOI:10.1007/s10236-013-0662-9*

---

## Author Comment (AC2) · 3 Sep 2019

**Reviewer # 2 Questions and our responses**

We thank Reviewer #2 for excellent comments and suggestions, which helped us to improve our paper. In this section, we first list the reviewer's question/comment, and then provide our answer. The questions/comments are in italics, and our response is in bold text.

*In this manuscript, an unsteady analytical solution was presented to simulate the spatial-temporal variation of salinity in convergent estuaries and applied to the Humen estuary of the Pearl River Delta. There are a lot of issues which should be addressed.*

*Major points:*
*1. This manuscript is about the unsteady state analytical model for salt intrusion, but in the introduction section there is no anything about unsteady state analytical model. Nobody else did the unsteady state analytical model?*

**At present, there are few studies presenting unsteady state analytical models to analyze the intratidal variation of salinity. Song et al. (2008) have proposed one applicable to laboratory flumes and rectangular canals, in a Chinese journal. We refer to this study in the introduction of the revised version, as below:**

**"…There are few studies focused on analyzing the intratidal variation of salinity analytically. Song et al. (2008) proposed an unsteady-state model applicable to laboratory flumes and artificial channels where the cross section is assumed to be constant along the channel. Here, an unsteady-state model is developed to predict the intratidal salinity intrusion dynamics in alluvial estuaries where the cross-section area typically converges."**

**Song, Z. Y., Huang, X. J., Zhang, H. G., Chen, X. Q., and Kong, J.: One–dimensional unsteady analytical solution of salinity intrusion in estuaries, China Ocean Eng., 22, 113–122, 2008.**

*2. What differences are there between your model and previous models? What are the advantages of your model? Authors should compare your model results with other model results, to prove that your model is better.*

**The unsteady analytical model developed by Song et al. (2008) can reproduce the salinity process in an idealized estuary with constant depth and constant width. Therefore, Song's model is best applicable to laboratory flumes and artificial channels. However, the convergence of cross-sectional area of estuarine channels is crucial. One innovation of this paper is to make use of the natural topography of alluvial estuaries, where the cross-sectional area development along the channel obeys an exponential function. So, our paper continues on Song's work within the geometrical setting of an alluvial estuary.**

*3. In the methods section, which are input parameters, and how to determine them? These should be presented clearly.*

**The input parameters include the tide-averaged salinity at the mouth, the convergence length of cross section $a$, the dispersion coefficient $D$, the tidal**

excursion $E_0$, the damping length of the tidal excursion $e$, the initial phase $\varphi_0$ and the tidal celerity $c$. We provide two approaches to estimate the calibrated input parameters. In the method section of the revision, we will introduce a way to calculate the tidal velocity $\upsilon$ (i.e. tidal excursion $E$) and the tidal propagation celerity ($c$) using the analytical hydrodynamics models by Cai, et al. (2012) and Cai and Savenije (2013). However, without geometry and friction data at the estuary mouth, the analytical model for tidal dynamics cannot be used in this case. Therefore, the input parameters in this study are calibrated using the measurements of salinity. The calibration of the parameters are presented one by one in Section 4.1 in the revised version.

Cai, H., Savenije, H. H. G., and Toffolon, M.: A new analytical framework for assessing the effect of sea-level rise and dredging on tidal damping in estuaries, J. Geophys. Res., 117, C09023, doi:10.1029/2012JC008000, 2012.
Cai, H., and Savenije, H. H. G.: Asymptotic behavior of tidal damping in alluvial estuaries, J. Geophys. Res., 118(11), 6107-6122, https://doi.org/10.1002/2013JC008772, 2013.

*4. In the application of the model to the Humen estuary, the first location of measurements (Dahu, figure 1) was set as the mouth of the estuary, and authors only calculated the results between station 1 and station 6 (figure 4). Actually, the real mouth is far downstream from station 1.*

The Humen estuary is the largest river outlet in Lingding Bay that connects the South China Sea and the Humen estuary. In this study, we choose the Dahu station (station 1) as the mouth of the estuary because it is usually considered as the bayhead of the Lingding Bay (Liu et al., 2000; Tian, 1986). The Dahu station is the connection point between the Humen estuary and Lingding Bay.

Liu, P., Wen, P., Zhou, Z., and Yu, T.: Analysis of influencing factor on shoal and though development of Lingdingyang Bay at Zhujiang Estuary, Journal of Oceanography in Taiwan Strait, 2000, 19(1), 119-124.
Tian, X.,: A study on turbidity maximum in Lingdingyang Estuary of the Pearl River, Tropic Oceanology, 1986, 2.

*5. In figure 2, the cross-sectional area of the Humen estuary was only shown for the reach between 0 km to 60 km. However, the Humen channel has a total length of 128 km (page 7, line 4). I think that the mouth in figure 2 should be the same as that in figure 4. If only part of the topography data was used, the area convergence length you obtained may be not correct. It is an important parameter in the model.*

Unfortunately, this is all the cross-section information we have at our disposal. We agree it would be better to use a longer stretch of the channel to estimate convergence length, but at the same time we have no reason to believe the channel geometry would not fit the same function in the part where we have no geometry data.

*6. Section 4.1 (Application to the Humen Estuary) is about calibration of model. Authors only discussed the calibration of parameters. The calibration results of model were shown in section 4.2 (model validation). In other words, model calibration and model validation used the same data. Although in figure 4 the results between 29 January and 3 February were shown, the conditions were similar.*

**We have rewritten Section 4.1 to clarify this in the revised version. The calibrated parameters include tidal excursion $E$ and dispersion coefficient $D$. Although each of the calibrated dispersion coefficient from 29 January to 3 February was listed in Table 2, in fact, only the one on 29 January was used in the study. In other words, we use the data on 29 January to calibrate the model parameters, and use the data from 30 January to 3 February to validate the model. We agree it would be interesting to see how the model performs under different conditions. This contribution can be considered a proof of concept. In the revised version, we use two figures to show the results, Figure 4 is the calibrated result and Figure 5 is the validation results, as below:**

[Figure]

**Figure 4: Comparison between calibration result and measured salinity concentration along the river on 29 January, 2005, showing values of measured salinity at high water slack (circle) and low water slack (inverted triangle), and the calibrated salinity curves at high water slack (red curve) and low water slack (blue curve).**

[Figure]

**Figure 5: Comparison between validation result and measured salinity concentration along the river from 30 January to 3 February, 2005.**

*7. Section 6 Conclusions. In this manuscript, the main work is application of the model to the Humen estuary, showing calibration results. The first paragraph is enough. In the second paragraph, part is about results instead of conclusions, and the other part is already in the first paragraph. In addition, "predictive", "predicating", and "predictable" used in conclusions are not proper.*

**We appreciate the reviewer's suggestion and deleted this part of conclusions in the revised version.**

*Minor points:*
*1. Page 1, lines 16-17: "Compared with steady-state solutions, it can directly reflect the influence of the tide and the interaction between the tide and runoff". Salt intrusion is the result of interaction between tide and runoff. The steady-state solution cannot reflect the influence of tide and interaction of tide and runoff? And authors did not compare their solution with steady-state solutions.*

**We agree that the steady-state solution can reflect tidal influence and interaction of the tidal motion and runoff. We have modified that inaccurate description in the revised version as below:**

**"…It is derived from a one-dimensional advection-diffusion equation for salinity, adopting a constant mixing coefficient and a single-frequency tidal wave, which can directly reflect the influence of the tidal motion and the interaction between the tide and runoff…"**

**There are two reasons why we did not compare our solution with other steady models in this paper. Firstly, in this study, we concentrated more on analyzing and discussing the ability of our unsteady model to capture the intratidal variation of the salinity. Secondly, the steady-state solution of our model obtained by**

integrating over the tidal period has the same expression as a widely used analytical model defined by Brockway et al. (2006). So, not surprisingly, our model applies well to the estimation of salinity distribution compared to the observations. Moreover, we did the relevant research and investigated the applicability of different steady solutions. Brockway's model has a simple calculation process and provides an accurate distribution of salinity in the downstream estuary (Xu et al., 2015).

Brockway, R., Bowers, D., Hoguane, A., Dove, V., and Vassele, V.: A note on salt intrusion in funnel–shaped estuaries: Application to the Incomati estuary, Mozambique, Estuarine Coastal Shelf Sci., 2006, 66, 1–5.
Xu, Y.W., Zhang, W., Chen, X.H., Zheng, J.H., Chen, X.W., Wu, H.X.: Comparison of Analytical Solutions for Salt Intrusion Applied to the Modaomen Estuary, J. Coastal Res., 2015, 31(3), 735-741.

*2. Page 1, line 31 and page 2, line 1: "Hence, the effects of human activities on salt intrusion are of major interest to engineers and scientists". This sentence is not related to the topic of this manuscript. Authors did not do anything about the influence of human activities.*

**We appreciate the reviewer's suggestion and deleted this sentence in the revised version.**

*3. Page 3, lines 2-5. The sentences about paper organization are not necessary.*

**Agreed, we deleted this part of introduction in the revised version.**

*4. Page 6. What is e in equations 17 and 18?*

**$e$ is the damping length of the tidal excursion. We explain this in the revised version as below:**

**"…where $E0$ is the tidal excursion at the mouth ($x=0$), and $e$ is the damping length of the tidal excursion…"**

*5. Page 6, line 20. Here the citation of a reference is not necessary. Particularly the reference is from a foreigner. Is a foreigner more familiar with a Chinese estuary?*

**We appreciate the reviewer's suggestion and deleted it in the revised version.**

*6. Page 7, line 6. What does the ES mean?*

**It was a mistake here. It should be "SE" which represents Southeast. We have corrected it in the revised version.**

*7. Page 7, lines 16-17: "The Humen waterway is well-mixed in the dry seasons (Luo et al., 2010)". The mixing condition can be seen directly from the vertical distribution of salinity, which should be shown in section 3.1 (overview of the study area).*

**The field survey was carried out by Guangdong Province Hydrology Bureau and the Pearl Hydrology Bureau from the River Conservancy Commission.**

Unfortunately, they only provided us the vertical averaged salinity at each measuring location because of the well-mixed condition in Humen estuary. To justify the lack of the vertical salinity data, we add more citations to support the assumption of well-mixed conditions in the revised version, as below:

"…The Humen estuary is well-mixed under normal flow conditions during the dry season (Ou, 2009; Luo et al., 2010). Due to three years of drought, the river discharge decreased by 50 percent during the study period in 2005 compared to a normal year (Liao, Pan, and Dong, 2008). Therefore, there is no doubt that well-mixed conditions prevailed during the calibration and validation…"

Liao, D.Y.; Pan, T.J., and Dong, Y.L., 2008. Characteristics of salt intrusion and its impact analysis in Guangzhou. Environment, S1, 4-5. (In Chinese)

Luo, L., Chen, J., Yang, W., and Wang, D.X, 2010. An intensive saltwater intrusion in the pearl river delta during the winter of 2007–2008, J. Trop. Oceanogr., 6, 22-28. (In Chinese)

Ou, S.Y., 2009. Spatial difference about activity of saline water intrusion in the Pearl River Delta. Scientia Geographica Sinica, 29(1), 89-92. (In Chinese)

*8. Page 7, section 3.2 data. What data about the tide was used in this study? In line 12, it is tidal flow. But in line 19, it is tidal level.*

The data of tidal flow is needed in our analytical solution, i.e. Eq (11). However, in this study, we used the tidal excursion instead of the tidal velocity, adopting a theoretical relation.

*9. The title of section 4.1 can be changed into "Model calibration", corresponding with section 4.2 Model validation.*

We appreciate the reviewer's suggestion and changed the title of section 4.1 into "Model calibration" in the revised version.

*10. Page 9, lines 18-19. Why did you use the daily maximum and minimum salinity in figure 4?*

Because the salinity were measured at hourly intervals. The daily maximum and minimum salinity were used as the approximate HWS and LWS salinity since the exact salinity values at HWS/LWS couldn't be obtained.

*11. Page 9, the last paragraph. I did not understand what authors wanted to express except for the first sentence. In the first sentence, the "downstream" is relative to the 40 km reach in figure 4 or the whole channel? It can be seen from figure 4 that the main overestimates occur at station 3 and station 5.*

Our study area is the downstream part of Humen estuary, therefore, "downstream" is relative to the whole channel. In Figure 4, we use 72 measured salinity observations at six stations at HWS and LWS to analyze the calculation results. As shown in Figure 4, overestimations occur at stations 2, 3, 4 and 5; 32 of the 72 measured salinity observations are overestimated compared with the

**calculation results, while 8 are underestimated.**

*12. Page 10, line 16: "salinity variation is more symmetrical further away from the study site". What does this sentence mean? It is difficult to understand.*

**The sentence means: Farther away from the mouth, the calculation of the intertidal variation improves, featuring more symmetry in the tidal cycle.**

*13. Page 10, lines 29-30. Authors used this sentence to explain the nonperiodic variation of salinity at Machong station in figure 5. It seems that only in the second tidal cycle, the variation is abnormal.*

**In comparison with the calculation results at the other stations, the model doesn't perform very well in Machong station. This may relate to nonperiodic variation in the velocity signal.**

*14. Page 16, table 2. All parameters used in the model should be shown.*

**We appreciate the reviewer's suggestion. All parameters used in the model are shown in Table 3 in the revised version as below:**

**Table 3 Calibrated values of Parameters**

| Parameter | Unit | Value |
|-----------|------|-------|
| $A_0$ | $m^2$ | 37822 |
| $a$ | km | 16.7 |
| $D$ | $m^2/s$ | 2562 |
| $E_0$ | km | 26.7 |
| $e$ | km | 30 |
| $c$ | m/s | 12 |
| $\varphi_0$ | rad/s | -0.7 |

*15. Page 17, figure 1.*
*(1) The Pearl River estuary is too complicated, and Humen is only one of eight branches. The figure caption is map of Humen estuary. But where is Humen? Only six gauging stations can be seen. The Humen estuary should be enlarged and shown clearly.*
*(2) River names "Beijiang River and Xijiang River" are different from the names "the North river and West river" in the text.*
*(3) East River and the Shiziyang channel in Page 10, line 23 were not shown in figure 1.*

**We appreciate the reviewer's suggestion and redraw Figure 1 in the revised version as below:**

[Figure]

**Figure 1: Map of the Humen estuary, showing the gauging stations where salinity concentration was measured during the field survey from 29 January to 3 February, 2005.**

*16. Page 19, caption of figure 3: "The linear relationship between these quantities predicted by Eq. (12) has been confirmed for all surveys, and the figures here show the linear line fitting results from Jan. 29th to Feb. 3rd". Page 22, caption of figure 6: "The subtidal discharge switches from seaward to landward between Machong and Dasheng stations, which will have an impact on salinity dynamics." These sentences should not be in the figure caption.*

**We appreciate the reviewer's suggestion and deleted them in the revised version.**

*17. The legends should be inside or outside figures, instead of covering the curves or words, such as figure 4, figure 6, and figure 7.*

**We redraw Figures 4, 6 and 7 in the revised version as below:**

[Figure]

**Figure 4: Comparison between calibration result and measured salinity concentration along the river on 29 January, 2005, showing values of measured salinity at high water slack (circle) and low water slack (inverted triangle), and the calibrated salinity curves at high water slack (red curve) and low water slack (blue curve).**

[Figure]

**Figure 5: Comparison between validation result and measured salinity concentration along the river from 30 January to 3 February, 2005.**

[Figure]

**Figure 11: Subtidal discharge measured at Machong station and Dasheng station from 29 January through 3 February. Positive values mean seaward.**

[Figure]

**Figure 12: Salinity and tidal flow velocity over a tidal cycle at Huangpuyou station. The measured salinity is represented by triangles and the measured flow velocity is indicated by circles (on 31 January 2005). The dashes line is the calculated tidal velocity while the dash-dotted line is the total velocity of tidal flow and river flow. The red solid curve represents salinity simulated by the unsteady analytical solution, which reproduces the time lag HWS and maximum salinity.**

*18. Is Humen a waterway or estuary? In some figure and table captions waterway was used, but in others estuary was used. It is the same in the text.*

**It should be Humen estuary. We have corrected the in the revised version.**

*19. English writing should be improved. For examples:*
*(1) Page 7, line 19, "salinity was obtained by using a salimeter". "by" or "using" is*

*enough.*

*(2) Page 9 and page 10. "Analysis of " in the titles of section 4.2.1 and 4.2.2 can be deleted. They are not necessary.*

*(3) Page 12, line 8, "the predicted result obtained by this model". "predicted" or "obtained" is enough.*

*(4) Page 16, caption of table 2: "Values of the parameters of salt intrusion in Humen estuary". "Values of the" can be deleted, "parameters" is enough.*

*These are only examples. Authors should check every sentence to make them standard, concise, and fluency*

**We appreciate the reviewer's suggestion and have made efforts to improve the English grammar in the revised version.**

---

## Editor Comment (EC1) · Hubert H.G. Savenije (Editor) · 5 Sep 2019

Dear authors,

I like the extensive reply that you gave to the reviewer. I just have a small question (maybe I missed something). But if I compare the equation (24) in the reply with equation (18) in the manuscript it seems that there is a difference of a factor 2. From a quick scan it seems to me that the factor 2 in (24) is not correct. Please can you shed some light on this issue?

256, 2019.

---

## Author Comment (AC3) · 5 Sep 2019

Dear Editor,

We thank you for your feedback. The question is in italics, and our response in bold text as below:

*I like the extensive reply that you gave to the reviewer. I just have a small question (maybe I missed something). But if I compare the equation (24) in the reply with equation (18) in the manuscript it seems that there is a difference of a factor 2. From a quick scan it seems to me that the factor 2 in (24) is not correct. Please can you shed some light on this issue?*

**Equation (14) in the manuscript is the analytic solution of the unsteady state salinity distribution, represented as:**

$$s = \overline{s}_0 \exp\left(\frac{Q_f a}{DA_0}\left(\exp\left(\frac{x}{a}\right)-1\right)\right)\left(1+\frac{E}{2a}\left(-\frac{Q_f a}{DA_0}\exp\left(\frac{x}{a}\right)\right)\sin(\omega t + \varphi)\right),$$  (14)

**where $\overline{s}_0$ is the tide-averaged salinity at the mouth of estuary, $D$ is the longitudinal dispersion coefficient, $A_0$ is the cross-sectional area at the mouth, $a$ is the convergence length of the cross-sectional area, and $E$ is the tidal excursion. Since the tide-averaged salinity along the estuary can be obtained as:**

$$\overline{s}_x = \overline{s}_0 \exp\left(\frac{Q_f a}{DA_0}\left(\exp\left(\frac{x}{a}\right)-1\right)\right),$$  (12)

**Equation (14) can be modified as:**

$$s = \overline{s}_x\left(1+\left(-\frac{EQ_f}{2DA_0\exp(-x/a)}\right)\sin(\omega t + \varphi)\right).$$  (S1)

**Introducing $A = A_0\exp(-x/a)$ into Equation (S1) yields:**

$$s = \overline{s}_x + \overline{s}_x \times \left(-\frac{EQ_f}{2DA}\right)\sin(\omega t + \varphi) = \overline{s}_x + \overline{s}_x \times I_s \sin(\omega t + \varphi),$$  (S2)

**where $I_s = -EQ_f/(2DA)$ was defined as the salinity amplitude coefficient, i.e. Equation (24) in the reply. Therefore, the factor 2 in Equation (24) is correct.**
**In addition, since the maximum salinity is reached at HWS and the minimum salinity is reached at LWS, Equation (S2) can be simplified for HWS into:**

$$s_{max} = \overline{s}_x + \overline{s}_x \times \left(-\frac{EQ_f}{2DA}\right),$$  (S3)

**and for LWS into:**

$$s_{min} = \overline{s}_x + \overline{s}_x \times \frac{EQ_f}{2DA}.$$  (S4)

**Thus combination of Equations (S3) and (S4) yields:**

$$\frac{s_{max} - s_{min}}{2} = \overline{s}_x \times \left( -\frac{EQ_f}{2DA} \right).$$

(S5)

**Rearrangement of Equation (S5) and using $x=0$ leads to the expression for the tidal excursion at the mouth, $E_0$:**

$$E_0 = \frac{\left( s_{max\,0} - s_{min\,0} \right) DA_0}{\overline{s}_0 \left( -Q_f \right)},$$

(S6)

**where $s_{max0}$ is the maximum salinity at the estuary mouth and $s_{min0}$ is the minimum salinity at the estuary mouth. Since the tidal excursion is assumed to decrease exponentially along the channel:**

$$E = E_0 \exp\left( -x/e \right),$$

(17)

**we can obtain Equation (18) in the manuscript by substitution of Equation (S6) in Equation (17):**

$$E = \frac{\left( s_{max\,0} - s_{min\,0} \right) DA_0}{\overline{s}_0 \left( -Q_f \right)} \exp\left( -x/e \right) = \frac{a\left( s_{max\,0} - s_{min\,0} \right)}{\overline{s}_0 \left( -\dfrac{aQ_f}{DA_0} \right)} \exp\left( -x/e \right).$$

(18)

**Therefore, there is no factor 2 in Equation (18) in the manuscript.**

Best regards,

Yanwen Xu, on behalf of all authors.

---

## Editor Comment (EC2) · Hubert H.G. Savenije (Editor) · 6 Sep 2019

Dear authors,

Thank you for this elaboration. I now see that eq.(24) is the amplitude, whereas eq.(18) refers to the range, which is twice as large. That was not immediately clear to me.

I shall now take the decision that you are invited to submit a revised paper and that in the supporting letter you clearly indicate the changes made. Since you have addressed the concerns of the reviewers so well in the discussion, I don't consider there is a need for a second review if these concerns are properly addressed in the revised paper.